# Implicit racial biases are lower in more populous more diverse and less segregated US cities

Andrew J. Stier [1,2] ✉, Sina Sajjadi [3,4], Fariba Karimi[3,5], Luís M. A. Bettencourt [6,7] & Marc G. Berman [1,8] ✉

Implicit biases - differential attitudes towards members of distinct groups - are pervasive in human societies and create inequities across many aspects of life. Recent research has revealed that implicit biases are generally driven by social contexts, but not whether they are systematically influenced by the ways that humans self-organize in cities. We leverage complex system modeling in the framework of urban scaling theory to predict differences in these biases between cities. Our model links spatial scales from city-wide infrastructure to individual psychology to predict that cities that are more populous, more diverse, and less segregated are less biased. We find empirical support for these predictions in U.S. cities with Implicit Association Test data spanning a decade from 2.7 million individuals and U.S. Census demographic data. Additionally, we find that changes in cities' social environments precede changes in implicit biases at short time-scales, but this relationship is bi-directional at longer time-scales. We conclude that the social organization of cities may influence the strength of these biases.

Cities are organized in surprisingly regular ways[1–3], which drive and constrain social interactions similarly across cultures and time[4–6]. However, there are many factors beyond the built-space geometry[2,3] of cities that modulate urban social interactions. Among these, implicit biases towards out-group members are one of the most universal[7]. Implicit biases refer to differential attitudes towards individuals from different groups, in ways that are automatic. These biases pose major barriers to equity and, in particular, implicit racial biases have been associated with disparities across essentially all aspects of life, including medical care[8], scholastic performance[9], employment[10], policing[11,12], mental health outcomes[13], and physical health[14]. If city organization and structure contribute meaningfully to these biases, there may be ways to leverage such regularities to systematically intervene and design for less biased urban areas. Despite the universality of implicit racial and ethnic biases in human

societies and their well-documented detrimental effects, existing studies lack a principled and theoretical basis to reveal how the organization of people in cities may systematically influence these biases.

Early investigations of the origins of implicit racial biases revealed that they develop early in life[15,16], are stable into adulthood, and are less prevalent in schools with more diverse populations[16]. Neurobiological evidence complemented these findings and showed that individuals with lower levels of bias process out-group stimuli more automatically. In particular, lower levels of implicit biases are associated with more automatic processing and less activation of a network of brain areas related to social context[17–20]. These observations suggested that early childhood exposure to diverse individuals is critical for building out-group expertize and locking-in low levels of implicit biases[21–23].

[1]Department of Psychology, University of Chicago, Chicago, IL, USA. [2]The Santa Fe Institute, Santa Fe, NM, USA. [3]Complexity Science Hub, Vienna, Austria. [4]Central European University, Vienna, Austria. [5]Graz University of Technology, Graz, Austria. [6]Department of Ecology and Evolution, University of Chicago, Chicago, IL, USA. [7]Mansueto Institute for Urban Innovation, University of Chicago, Chicago, IL, USA. [8]The University of Chicago Neuroscience Institute, University of Chicago, Chicago, IL, USA. ✉e-mail: stier@santafe.edu; bermanm@uchicago.edu

However, more recent work has demonstrated that interventions with older children and adults that increase exposure to out-group individuals also reduce implicit biases, although these effects wear off if the intervention is not continued[24–26]. This suggests that individuals' biases likely reflect ongoing predictions about their social environment[27,28], and consequently, that consistent population averages of implicit biases[29] are the result of consistent social contexts. Thus, earlier findings of stable implicit biases throughout adulthood likely reflect, in fact, not stable individual cognitive biases but instead the stability of social environments[27–30].

For example, the effects of slavery and associated racial segregation in the United States (U.S.) on social context and network structure have been enduring. Areas in the U.S. with larger slave populations in 1860 have higher current levels of implicit racial biases today[30]. This example demonstrates one way in which longstanding structural influences on social contexts (e.g., racism) may contribute to implicit biases and perpetuate them across generations. Given the strong influence of city organization on urban social interactions and contexts[1,2,31], it is natural to ask if there are general ways in which urban environments might shape implicit biases.

In this work, we begin to answer this question by developing a mathematical model linking the properties of cities with implicit biases. This model specifies learning as the mechanism linking properties of the urban environment to biases and is inspired by previous urban science, psychological, and neurobiological research. The model predicts that larger, more diverse, and less segregated cities have lower levels of implicit racial biases. We find that this prediction is consistent with implicit association test data from 2.7 million individuals over ten years and we discuss additional predictions of our model. Note that these results do not provide direct causal confirmation for the proposed mechanism.

## Results

We start our analysis of urban composition from the point of view of urban scaling theory[1,2]. Its mathematical models describe cities as social networks enabled and structured by cities' hierarchical infrastructure networks. In this type of model, cities arise as the result of balancing the spatial costs of housing and the transportation of goods and people with the benefits of facilitating social interactions over cities' infrastructure networks[1,2]. These models derive average properties of cities as a function of their population size, $N$, as scale-invariant scaling relationships[1,2]. For example, in the case of average per-capita social interactions, $k$, the scaling relationship takes the form of $k \sim N^{\delta}$, where $\delta = \frac{1}{6}$. Here, scale-invariance refers to the property that doubling $N$ results in a $2^{\delta} - 1 \approx 12\%$ increase in per-capita social interactions, $k$, regardless of initial values for $N$ and $k$.

In the simplest models of urban scaling theory, all urban inhabitants are taken to be equally likely to interact (i.e, there is homogeneous mixing) and all inhabitants are treated, in this sense, identically. In our related work, we developed modifications of these models to account for individuals belonging to distinct groups and for the fact that their connections may be biased by group identities, such that individuals may interact less often with out-group individuals and more often with their in-group[32]. This translates into groups that may show increased same-group interaction tendencies (homophily) or decreased between-group interaction tendencies. In developing this model, our focus was on understanding how homophily, segregation, and group sizes impact emergent socio-economic outputs in cities as a result of the inhibition of a number of interactions across individuals of different racial and ethnic groups. However, here, we focus more directly on what this model can reveal about systematic variations in inter-group interactions and subsequent consequences for implicit biases.

The model of heterogeneous group interaction describes the number of per-capita interactions, $k_i$ in city $i$, on average, as:

$$k_i \sim N_i^{\delta} \left[ \sum_{g=1}^{G} \left( \frac{N_{g,i}}{N_i} \right)^2 (1 + h_{g,i}^{in}) + \sum_{g=1}^{G} \sum_{j \neq g} \frac{N_{g,i}}{N_i} \frac{N_{j,i}}{N_i} (1 - h_{g,i}^{bet}) \right] \quad (1)$$

Here, $g$ indexes the distinct groups in cities, $h_{g,i}^{bet}$ and $h_{g,i}^{in}$ are the between-group and within-group relative rates of interactions of group $g$ in city $i$, and $N_{g,i}$ is the population of group $g$ in city $i$. In this model, individuals from group $g$ in city $i$ interact with out-group individuals with a relative rate $1 - h_{g,i}^{bet}$ and with in-group of $1 + h_{g,i}^{in}$[32]. In addition, we have made the assumption that each group avoids all other groups similarly so that there are no unique avoidance effects between pairs of groups[32].

The first term of Equation (1) is the typical scaling relationship[1,2,4]. The second term has two components, each representing fractions of the total number of possible social interactions, $N^2$. The first of these captures social interactions which occur within groups, on average: $k_{within,i} \sim N_i^{\delta} \cdot \sum_{g=1}^{G} \left( \frac{N_{g,i}}{N_i} \right)^2 (1 + h_{g,i}^{in})$. The second term captures social interactions which occur between groups, on average: $k_{inter,i} \sim N_i^{\delta} \cdot \sum_{g=1}^{G} \sum_{j \neq g} \frac{N_{g,i}}{N_i} \frac{N_{j,i}}{N_i} (1 - h_{g,i}^{bet})$. Though there is some evidence that high levels of implicit biases are associated with increased homophilic tendencies[33,34], these studies do not discuss alternative mechanisms for inducing changes in implicit biases other than changes in inter-group interactions resulting from homophily (see Supplementary Note and Supplementary Figs. 1 and 2). In contrast, there is a large body of previous research that has qualitatively demonstrated that inter-group interactions shape implicit racial biases[16,24,27,28,35–39]. Thus, we focus on this term to build our model.

In order to explicitly connect the quantity of inter-group interactions in cities to implicit bias levels, an additional step is required to translate from inter-group interactions to levels of implicit biases[1]. Previous research has suggested that this relationship is positive – more inter-group interactions are associated with lower implicit bias levels[16,24,27,28,35–39]. In addition, neurobiological studies provide evidence that individuals with lower levels of bias engage in more automatic processing of out-group stimuli, indicating greater expertize[17–20].

A common feature of such expertize-based learning is decreasing marginal returns to exposure, which is often formalized in a learning curve[40–43]. Learning curves describe the relationship between costs and expertise across diverse individual or group tasks such as motor learning[41], sequence learning[42], solar panel construction[43], and cigar rolling[40]. Typically, these learning curves are described by power-laws of the form $cost \sim n^{-\alpha}$, where $n$ is the number of learning instances, and $1 > \alpha > 0$ determines the speed of learning (or learning rate, $\alpha = - d \ln cost / d \ln n$), with larger values of $\alpha$ implying faster learning.

Such learning curves are a natural modeling choice to couple inter-group interactions and implicit bias levels since our measure of implicit bias, $b$, can be interpreted as a cognitive processing cost: $b$ is a relative difference in reaction times when pairing photographs of White and Black faces with positive and negative words, see Methods. Thus, decreasing $b$ can be seen in this context as learning that increases social performance in a diverse population, and such learning is the result of greater levels of exposure (interactions) to out-group individuals.

With the additional assumption that coupling strength and direction do not vary between different pairs of groups or across interaction types (e.g., friendship, employment, acquaintance, etc)[1,2], we expect measured bias levels to follow a learning curve of $b_i \sim k_{inter,i}^{-\alpha}$ and therefore, we predict larger cities systematically have lower levels of bias according to:

$$b_i \sim N_i^{-\delta\alpha} \cdot \left[ \sum_{g=1}^{G} \sum_{j \neq g} \frac{N_{g,i}}{N_i} \frac{N_{j,i}}{N_i} (1 - h_{g,i}^{bet}) \right]^{-\alpha} \quad (2)$$

In the presence of reduced between-group interactions ($h_{g,i}^{bet} \neq 0$), it is interesting to consider the case of cities with only two distinct groups. This approximation is particularly relevant to the measure of implicit racial bias we employ here which explicitly contrasts White and Black racial groups. In this case, the scaling relationship for implicit racial biases simplifies to (see Supplementary Note):

$$b_i \sim N_i^{-\delta\alpha} \cdot \left[ \frac{N_{1,i}}{N_i} - \left( \frac{N_{1,i}}{N_i} \right)^2 \right]^{-\alpha} \cdot (2 - h_{1,i}^{bet} - h_{2,i}^{bet})^{-\alpha} \qquad (3)$$

Equation (3) can be understood in terms of three multiplicative terms: a scaling relationship, a diversity adjustment, and a segregation adjustment. Inter-group interactions drop dramatically as diversity decreases and less dramatically as the segregation values of the groups increase (see Methods). In practice, since some cities are not very diverse ($\frac{N_1}{N} \sim 1$) and segregation values are small (Supplementary Fig. 4), diversity is expected to play a much larger role than segregation in determining the average number of inter-group interactions and in driving subsequent implicit biases.

In addition, Equation (3) also predicts that the logarithms of the diversity adjustment and the segregation adjustment should be negatively and linearly related to the logarithm of implicit bias, $b$. These two adjustment terms capture deviations from the mean-field scaling relationship ($b \sim N^{-\delta\alpha}$) due to the specific characteristics of each given city. In summary, the model predicts that larger, more diverse, and less segregated cities have lower average levels of implicit biases.

Finally, the model suggests that deviations of the scaling exponent away from $\delta = \frac{1}{6}$ and the magnitude of the diversity effect can provide empirical estimates of the learning rate, $\alpha$, which characterizes the coupling between inter-group interactions and implicit racial biases. Since values of reduced between-group interactions are not directly observed (see Methods), we cannot obtain a direct estimate of $\alpha$ from the third term of Equation (3). In addition, we note that there may be other sources of deviations from the expected scaling exponent of $\delta = \frac{1}{6}$ including top-down hierarchical constraints on intergroup interactions[44], growth rate fluctuations, and other higher-order effects[45], which may contribute to differences in independent estimates of $\alpha$ calculated from the first and the second terms of Equation (3).

We next test the three predictions of our model: (1) that implicit biases systematically decrease with city size via a scaling relationship of $b \sim N^{-\delta\alpha}$, (2) that cities with more diversity have lower levels of implicit biases, and, (3) that less segregated cities have lower levels of implicit biases.

We used data from the racial Implicit Association Test (IAT) to quantify the level of implicit racial bias in U.S. cities for each year in 2010-2020[46]. The racial IAT measures the difference in response times when subjects pair images of White versus Black faces with positive or negative words (Fig. 1). We linked average IAT bias scores from approximately 2.7 million individuals in combined statistical areas (CBSAs) with racial demographics and population data from the U.S. Census to test our predictions. We note that CBSAs are functional definitions that capture the spatiotemporally extended social networks of cities and include, in the same unit, where people live, socialize, and work[47].

In addition, it is important to note that this sample is not nationally representative and tends to be younger, more educated, with a higher percentage of female participants, and likely underestimates bias levels, overall[48]. Nonetheless, racial demographics are strongly correlated across cities suggesting that this sample is suitable for relative comparisons across cities (Spearman correlation, $r_s \in [0.83, 0.93]$ for the White population/IAT sample fraction and $r_s \in [0.93, 0.96]$ for the Black population/IAT sample fraction; Supplementary Table 1 and Supplementary Fig. 3; note that complete model results and statistics for all models are available in Supplementary Data 1).

We measured reduced between-group interaction values, $h_i^{bet}$, as linearly dependent (see Supplementary Fig. 5 for equivalent

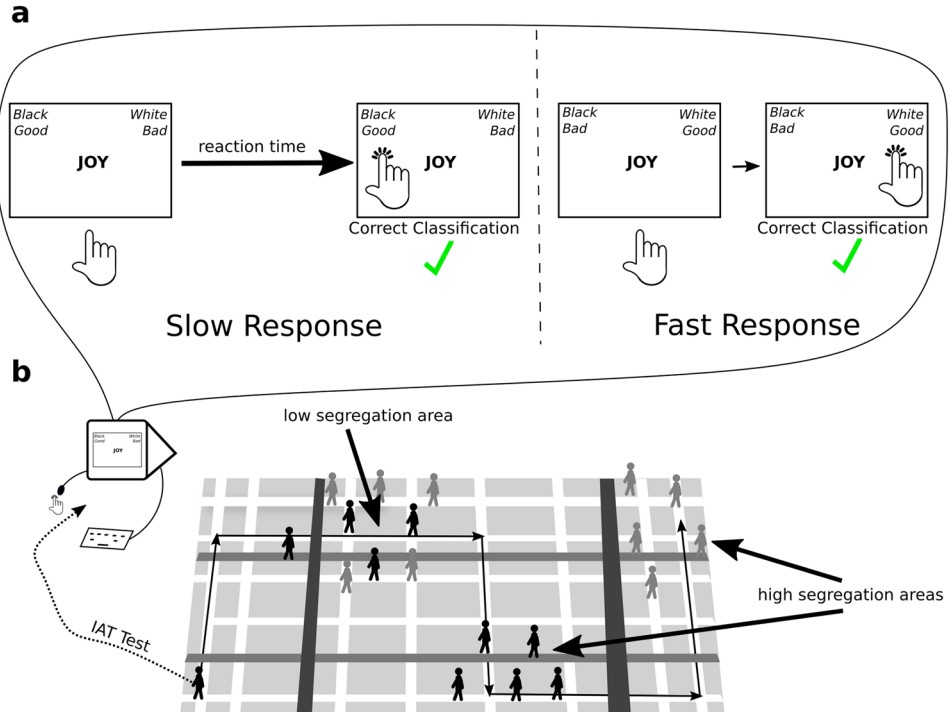

**Fig. 1 | A schematic depiction of the Implicit Association Test (IAT) and our model. a** The IAT measures implicit racial biases as a relative difference in reaction times between different pairings of word and face categories. **b** We model implicit racial biases in cities as a cumulative exposure process to out-group individuals shaped by city population size, demographic diversity, and residential racial segregation.

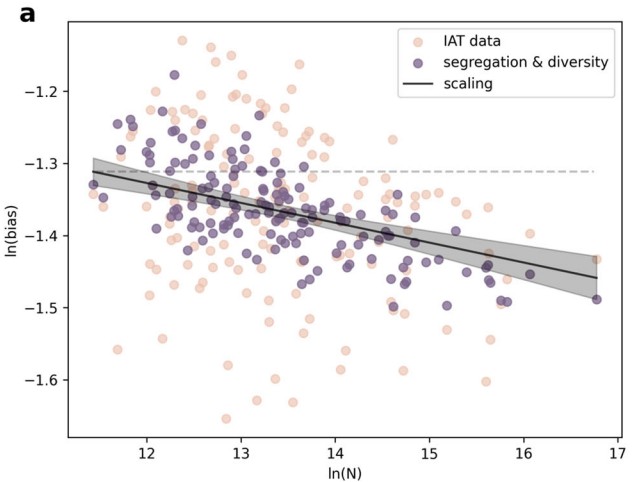

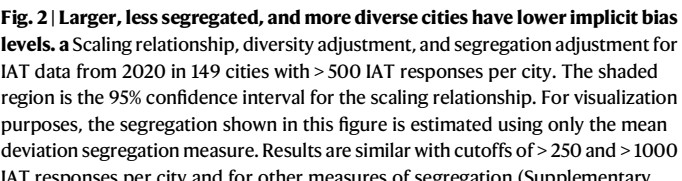

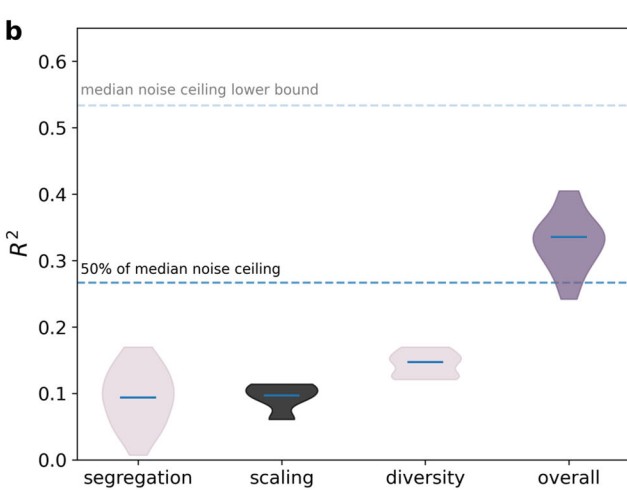

**Fig. 2 | Larger, less segregated, and more diverse cities have lower implicit bias levels. a** Scaling relationship, diversity adjustment, and segregation adjustment for IAT data from 2020 in 149 cities with > 500 IAT responses per city. The shaded region is the 95% confidence interval for the scaling relationship. For visualization purposes, the segregation shown in this figure is estimated using only the mean deviation segregation measure. Results are similar with cutoffs of > 250 and > 1000 IAT responses per city and for other measures of segregation (Supplementary

Tables 19–26). **b** Variance explained ($R^2$) by segregation (measured via residential racial segregation), diversity, and scaling relationship. Data for $n = 20$ models are shown for 2016–2020. Medians are shown by a horizontal line and have values of 0.094, 0.097, 0.147, and 0.346, respectively. Variance explained by segregation is from all four models with different segregation measures. Noise ceiling estimates are obtained by computing correlations of bias levels between split halves of IAT participants within cities.

analyzes with a non-linear dependence) on residential racial segregation calculated from racial demographics in census tracts (small areas of ~ 4,000 inhabitants). The choice to proxy these values with segregation measures is motivated by past empirical[49,50] and theoretical work (e.g.,[51,52]) linking population mixing and segregation. We repeated this statistical analysis across four distinct measures of residential racial segregation, as in our related work[32]. We find that across all years and measures of residential racial segregation, larger cities have lower levels of implicit racial biases, in line with Equation (3) (95% confidence interval for the population coefficient: $\beta_1 \in [-0.045, -0.031]$; Fig. 2a, Supplementary Table 2).

In addition, more diversity and higher levels of residential racial segregation are significantly related to scaling deviations and associated with higher average IAT scores, in line with Equation (3) (95% confidence intervals for the diversity and segregation coefficients: $\beta_2 \in [-0.226, -0.163]$, $\beta_3 \in [0.026, 0.066]$; Supplementary Table 2). Importantly, the diversity and segregation terms can be statistically separated even though they are correlated (maximum variance inflation factor of 6.31 across all four segregation measures; Supplementary Fig. 6). We note that when analyzing single years of data before 2015, residential racial segregation is not significantly related to scaling deviations for some segregation measures. However, this is likely due to much lower sample sizes in those years resulting in fewer cities with available data and smaller fractions of city populations represented (average percent of city population before 2015: 0.078%; average percent of city population after 2014: 0.168%; Supplementary Table 3; see Supplementary Data 2 for a list of cities included in each year).

Further, the city size scaling, diversity, and residential racial segregation effects are predictive of individual IAT responses when controlling for race, birth-sex, and educational attainment (population coefficient range $\beta_1 \in [-0.0404, -0.0124]$, diversity coefficient range $\beta_2 \in [-0.1155, -0.1937]$, segregation coefficient range $\beta_3 \in [0.1951, 0.7081]$; Supplementary Tables 4–14; note that the coefficients on diversity were only significant after 2015). This suggests that these large-scale structural determinants of implicit racial biases are relevant to individuals' levels of bias. In other words, citywide organizational and structural characteristics may influence individual implicit biases despite the diversity of local social environments (e.g., variation in

neighborhood segregation compared to city-wide averages) that any individual urban inhabitant might encounter.

Along these lines, other research has identified environmental variables related to area deprivation associated with inter-city variance in implicit racial bias[53]. However, with our model, we find that measures of area deprivation independently explain only a small portion of the variance in inter-city differences above and beyond the three structural factors we identify here (Supplementary Tables 15–18). This suggests that the variables identified previously actually capture a combination of city population, segregation, and diversity (e.g., see Supplementary Fig. 7) and that there are other factors, for example, segregated mixing in ambient populations[54], that may explain the remaining inter-city variance in implicit biases.

In addition, we observe that for 2015-2020, systematic variations in city size, diversity, and segregation account for a median of 33.6% (with a range of [24.2%, 40.5%]) of the variance in implicit racial bias across cities (and all four segregation measures), which is equivalent to a correlation of $r \sim 0.58$ (range of $r \sim [0.49, 0.64]$, Fig. 2B, Supplementary Tables 27–33).

In order to better understand the performance of our model, we employ estimates of the noise ceiling[55,56]. Since implicit biases are inherently noisy attitudes[28], meaning that they fluctuate frequently, a model that is perfectly predictive may still fail to explain all of the observed variance and have an $R^2 < 1$. Noise ceiling estimates provide the maximum $R^2$ value that can be expected, given the level of noise in the data. Here, these estimates suggest that the three structural factors in our model capture a majority of the variance that can be accounted for given the reliability[57] of the IAT measure (noise corrected $R^2$ range $\in [0.38, 0.93]$; Supplementary Tables 27–36). As expected, based on the fact that many U.S. cities are not so diverse, diversity accounts for more between-city variance in implicit biases than residential racial segregation (diversity $R^2 = .16$, segregation $R^2$ range $\in [0.008, 0.082]$ including all years of data; Fig. 2b, Supplementary Tables 27–33).

Finally, we compared estimates of the learning rate, $\alpha$, to previously conducted experimental interventions[25,26] designed to simulate inter-group contact. The two independent estimates of $\alpha$, from the scaling exponent and the diversity adjustment (see Methods), are convergent and consistent (Fig. 3). This need not have been the case

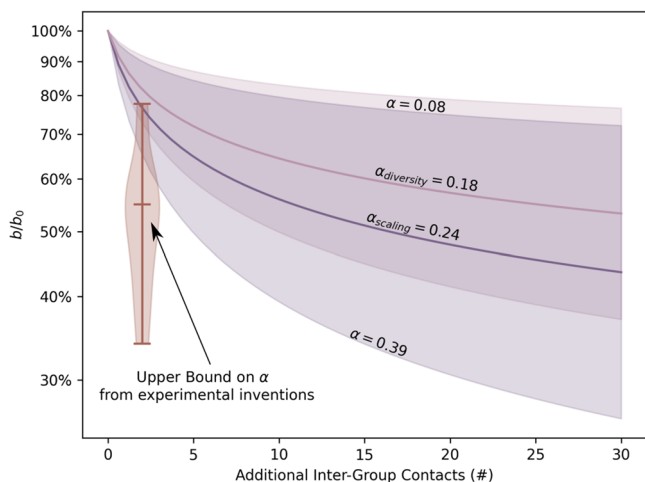

**Fig. 3 | Estimated learning rates, α.** We plot learning as a decrease in bias levels relative to an arbitrary baseline, $\frac{b}{b_0}$ as a function of the number of additional inter-group contacts. Solid curves indicate the mean estimated learning rate from the scaling exponent or majority group adjustment (diversity effect) averaged across years. Shaded regions show the 95% confidence intervals for the learning rate estimates with the lower envelop and upper envelope referring to the scaling exponent and diversity estimates, respectively. The violin plot gives an upper bound on the learning rate from 18 previously conducted experimental interventions[25,26] designed to simulate one-shot inter-group contact of varying quality.

and this convergence of estimates provides empirical support for a shared mechanism (namely a learning curve as a function of out-group exposure) coupling city population and diversity to implicit bias levels. These empirical estimates of the learning rate are also consistent with experimental interventions – in which simulated inter-group contact is overwhelmingly positive and occurs immediately before bias measurements – that provide an upper bound on the learning rate, α (see Methods). These results suggest that observed levels of implicit biases emerge from the interaction between large-scale structural factors operating across entire cities to shape social contexts, and individual psychology which determines how much and how quickly people learn from and internalize those social contexts.

### Timescales of temporal precedence

The learning mechanism linking biases and between-group interactions emphasizes a specific causal direction in the model: interactions → bias levels. However, there are other mechanisms, such as selective migration[58] and individual mixing preferences, that may facilitate reverse causal pathways in which bias levels influence changes in diversity, population size, and segregation, respectively. While the between-group interaction term in the model can account for the effects of mixing preferences (along with historical processes and explicit racism, e.g., that influenced unfair lending policies), our model does not explicitly account for processes in which implicit biases facilitate changes in city diversity and population size.

To begin to understand the role of each of these causal directions, we take advantage of the fact that 43 cities have implicit racial bias data available for all 10 years. We employ Granger causality[59] to statistically test whether changes in one variable precede or follow changes in another variable. In brief, these analyzes test whether the linear regressions between two variables of interest improve when one of the variables is lagged in time (see Methods). We perform these analyzes for each city and calculate the percentage of cities with statistically significant evidence of temporal precedence.

We find evidence that changes in population size, diversity, and segregation precede changes in implicit biases at a lag of one year for a majority of cities (Table 1, Fig. 4). In contrast, only a fraction of cities

**Table 1 | Percentage of 43 cities with evidence for a given temporal precedence direction**

|  | 1 year lag | 2 year lag | 3 year lag |
|---|---|---|---|
| population → bias | 73.8 ± 7.0 | 78.6 ± 6.3 | 85.7 ± 5.5 |
| bias → population | 19.0 ± 5.9 | 35.7 ± 7.5 | 76.2 ± 6.2 |
| diversity → bias | 61.9 ± 7.3 | 66.7 ± 7.3 | 88.7 ± 4.9 |
| bias → diversity | 24.4 ± 6.7 | 45.2 ± 7.7 | 84.5 ± 5.6 |
| segregation → bias | 69.0 ± 7.1 | 76.2 ± 6.7 | 95.2 ± 3.3 |
| bias → segregation | 19.0 ± 6.3 | 38.1 ± 7.5 | 81.0 ± 5.9 |

Data are presented as means with errors representing the bootstrapped standard deviation of the mean. A two-tailed sum of squared residuals $\chi^2$ test was used to determine statistical significance.

show evidence for the reverse temporal precedence. Results are similar at a lag of 2 years. At a lag of 3 years, however, there is equal evidence for both temporal precedence directions.

In combination with the mathematical model presented here, these results suggest a mismatch in the timescales at which different mechanisms play out. In particular, these analyzes suggest that, at short timescales (i.e., 1–2 years), changes in structural factors primarily precede changes in implicit racial biases as individuals learn from and internalize changing social contexts. However, there is also some evidence of the reverse temporal direction at these short timescales. This direction, of bias changes preceding structural factors, may be due to immediate, individual-level effects such as changes in bias levels leading to changes to individual mixing preferences.

At long timescales, evidence is present for influence in both directions from biases to structural factors and vice versa. This fits with our model's suggestion of rapid learning involved in setting implicit biases (Fig. 3). Psychological adaptations to changing social conditions are expected to be faster than the speed with which individuals (and their households) can move to different neighborhoods or cities. Thus, we expect that changes in biases happen faster than changes in city demographics and patterns of segregation. More work is needed to enumerate potential mechanisms linking bias levels back to structural changes (e.g., selective migration[58]) and to mathematically model these mechanisms in an urban scaling context.

### Discussion

The model developed here demonstrates that relatively simple considerations of heterogeneous mixing among a small number of social groups can explain a large proportion of why people in some cities have stronger implicit racial biases than in others. While it is somewhat surprising that only three factors - city population, diversity, and racial segregation - account for so much between-city difference, this is in line with recent evidence that implicit racial biases are driven more by social contexts than by individual differences in attitudes[25,26,60,61].

Our model provides a number of concrete theoretical predictions that may form the basis of new experimental hypotheses. First, our model predicts that at short timescales, implicit racial biases emerge from the interaction between city-wide social contexts that are shaped by the built environment and individual psychology which determines how much and how quickly people learn from those contexts. We find preliminary support for this hypothesis by taking advantage of the longitudinal nature of our data (Fig. 4, Table 1). At longer timescales, other still unenumerated, and unmodeled mechanisms may create feedback loops in which implicit racial biases shape these social contexts, e.g., through selective migration[58].

Second, our model implicitly predicts that on average, inter-group contact in cities is beneficial with respect to reducing implicit racial biases. This matches results from the urban scaling literature that includes psychological depression[62], economic outputs[1,2,32], and

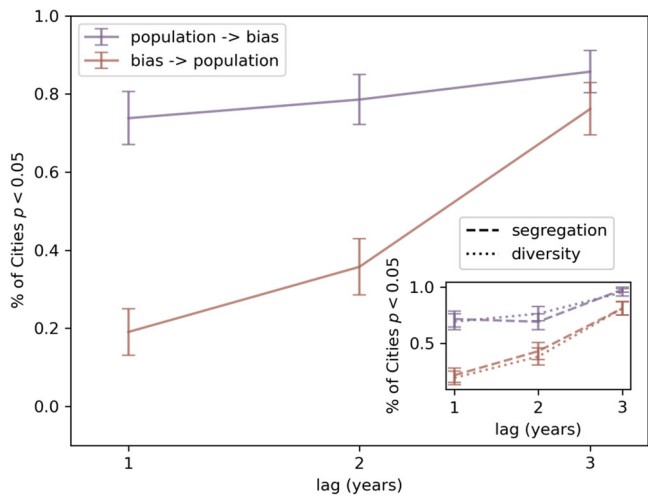

**Fig. 4 | Granger causality analyzes provide differing amounts of evidence for each direction temporal precedence across 43 cities.** At lags of one and two years, more cities have evidence of changes in population preceding changes in bias. At a lag of three years, there is equal evidence for both directions. Data are presented as means with error bars represent the bootstrapped standard deviation of the mean. The inset shows the same measure for diversity (dotted line) and segregation (dashed line) with Granger causality directions indicated by the same colors. A two-tailed sum of squared residuals $\chi^2$ test was used to determine statistical significance.

creative outputs[63]. In all of these cases, the observation of increasing beneficial returns to city population suggests that interactions across these modalities are, on average, positive. If this was not the case, we would expect to find all three main results reversed so that smaller, less diverse, and more segregated cities have lower bias levels. This was not what we found empirically.

However, the equations of urban scaling theory as formulated do not address interaction quality directly. There is likely great variation in interaction quality within cities. For example, inter-group contact may be cognitively costly[64], and interactions between individuals or in certain neighborhoods may be negative, particularly in areas with high levels of existing implicit racial biases[65]. Thus, investigations of whether and how cities systematically facilitate interactions of differing quality are natural next steps.

Finally, our model predicts that as more people move into cities over the next decades implicit biases may decrease so long as cities are not too segregated, remain centers of diversity, and residents learn from shifting social environments. In addition, our model predicts that decreasing segregation may lead to reductions in implicit racial biases that could have large societal impacts[66], though causal evidence is needed to confirm these hypotheses. Such reductions in segregation may have implications beyond implicit biases as cities with lower levels of racial segregation also tend to have higher incomes[32] and healthier inhabitants[67].

In summary, these results, along with our related work[32] characterizing economic productivity, are first steps towards better incorporating heterogeneous network structures and individual psychology into the mathematical models of modern urban science and deriving associated multifaceted effects. The additions we developed here are relatively simplistic in their consideration of individual differences in cities, proxied simply by a set of discrete groups. More complex models are likely needed to consider how city organization influences the dynamics of other types of attitudes that are socially relevant, including political polarization[68,69] and issues of trust and collective action, for example relating to public health programs such as vaccines[70,71].

## Methods

### IAT Data
All racial IAT Data are publicly available[46] and were downloaded from https://osf.io/52qxl/. The collection of these data was approved by the University of Virginia Institutional Review Board for the Social and Behavioral Sciences. The use of these data in the present study was approved by the University of Chicago Social Sciences Institutional Review Board (IRB23-0796). These data are coded at the participant level, a fraction of which includes geographic identifiers for state and county. Implicit racial bias was assessed by the $D_{biep}$ metric[72] which consists of the latency difference between compatible and incompatible blocks of the racial IAT, divided by the pooled standard deviation. In the racial IAT, Black and White face images are used and higher and positive $D_{biep}$ scores indicate an implicit bias towards White faces while lower and negative $D_{biep}$ scores indicate an implicit bias towards Black faces. After only retaining participants with available geographic information, $D_{biep}$ scores were averaged across all participants in each CBSA. Cities were retained if they had at least 500 IAT responses. This was done separately for all years. Results were similar with cutoffs of > 250 and > 1000 IAT responses per city (Supplementary Tables 19–26). We note that multiple comparison corrections are not relevant to these various robustness checks or to the various versions of models run with data from different years and different segregation measures: these test the same hypothesis on independent datasets rather than testing/comparing multiple hypotheses within the same dataset.

### U.S. census data
All census data is publicly available and was downloaded from data.census.gov. Five-year racial demographic estimates for U.S. census tracts were downloaded from table B02001. segregation values were calculated across the two racial groups in the race IAT: White and Black. Five-year population estimates for U.S. cities defined as combined statistical areas (CBSAs) were downloaded from table B01003. In order to map between census tracts and CBSAs, delineation files for 2020 were downloaded from the United States Office of Budget and Management from https://www.census.gov/programs-surveys/metro-micro/about/delineation-files.html.

### Associations between implicit bias, city size, diversity, and segregation
We fit the scaling relationship between the logarithms of implicit bias and city size with ordinary least squares (OLS) linear regression to determine the scaling exponent. The equation for this regression is:

$$\ln(b_i) \sim C + \beta_1 \cdot \ln(N_i) + \epsilon_i \tag{4}$$

where $C$ is the log-log intercept (or equivalently the logarithm of the scaling prefactor), $\beta_1$ log-log slope (i.e., the scaling exponent), and $\epsilon_i$ are the scaling deviations.

In order to assess the contribution of the city-specific diversity and segregation values to implicit racial bias, we start with $\epsilon_i$ as the dependent variable via the equation:

$$\epsilon_i \sim C_2 + \beta_2 \cdot \ln\left(\frac{N_{1,i}}{N_i} - \frac{N_{1,i}^2}{N_i^2}\right) + \beta_3 \cdot \ln(2 - h_{1,i} - h_{2,i}) + \xi_i \tag{5}$$

where $N_{1,i}$ is the number of White individuals city $i$, $h_{1,i}$ is the segregation of the White population, and $h_{2,i}$ is the segregation of the Black population, and $\xi_i$ are additional city specific residual effects.

Since we do not observe segregation values, $h_{1,i}$ and $h_{2,i}$, directly, but only measures of residential racial segregation, $s_{1,i}$ and $s_{2,i}$, we follow our related work[32] and model the segregation values as linearly

dependent on levels of residential racial segregation. With the additional approximation that $\ln(2 - x) \simeq \ln(2) - \frac{x}{2}$ when $x \ll 1$, equation (5) then becomes:

$$\epsilon_i \simeq C_2 + \beta_2 \cdot \ln\left(\frac{N_{1,i}}{N_i} - \frac{N_{1,i}^2}{N_i^2}\right) - \frac{\beta_3}{2} \cdot [2 \cdot h^{bet} + b^{bet} \cdot (s_{1,i} + s_{2,i})] + \beta_3 \ln(2) + \xi_i$$

(6)

where we have substituted the segregation values via the equation $h_{g,i} = h^{bet} + b^{bet} s_{g,i}$[32]. We can further simplify by including all non-city specific effects in the constant $C_2$ and by including the factor of $\frac{-b^{bet}}{2}$ in the constant, $\beta_3$. We fit the resulting equation via OLS in order to assess the contribution of diversity and residential racial segregation to implicit racial bias:

$$\epsilon_i \simeq C_2 + \beta_2 \cdot \ln\left(\frac{N_{1,i}}{N_i} - \frac{N_{1,i}^2}{N_i^2}\right) + \beta_3 \cdot (s_{1,i} + s_{2,i}) + \xi_i$$

(7)

## Noise ceiling estimates

To better understand the performance of our model, we computed the bounds of the noise ceiling for the implicit bias measure. The idea of a noise ceiling is borrowed from cognitive neuroscience[55,56] where the performance of predictive models can be limited by inherent noise in brain activity and measurement noise from human neuroimaging devices (e.g., functional magnetic resonance imaging or electroencephalogram). In those settings, a perfectly predictive model would only explain a fraction of the variance observed in the data (i.e., $R^2 < 1$). Therefore, without an estimate of the noise ceiling, it is impossible to assess whether a model fails to reach an $R^2$ close to 1 due to limitations in the model or the underlying measurement. This concept is not specific to brain imaging and can be applied to any measurement that is known to be noisy. Using noise ceiling estimates to evaluate models of implicit biases is appropriate because implicit biases are high entropy attitudes[73] and hence inherently more difficult to measure.

In order to estimate the noise ceiling, we computed the correlation between IAT bias measures between halves for 500 split permutations of individual IAT participants in each year. The upper bound of the noise ceiling was estimated by averaging the correlations between each half and the full sample, while the lower bound of the noise ceiling was estimated by correlating IAT bias between the two halves of each split half[55,56].

## Measures of residential segregation

As in our related work[32], all analyzes were conducted across four different measures of residential segregation[74] in order to ensure that the results were not sensitive to any specific metric. Each of these measures has its own drawbacks and benefits. Each one differs with respect to how changes in the spatial distribution of the population affect the measure and how the measure behaves with respect to an uneven distribution of population throughout the city.

These measures included the mean deviance measure:

$$\Delta_{g,i} = \frac{1}{M} \sum_m^M |N_{g,m,i}/N_{m,i} - N_{g,i}/N_i|,$$

(8)

with $g$ indexing group, $m$ indexing neighborhood, and $i$ indexing city. This can be interpreted as the percentage of each group that would have to change residences to produce an even distribution throughout a city. However, the movement of people between neighborhoods that are above the mean for that group does not change this measure. In other words, the movement of individuals between two neighborhoods that have a higher percentage of Black (or White) residents than the city as a whole will not impact this measure. In addition, this

measure does not account for cases in which some neighborhoods have a much larger share of the population.

The normalized segregation index:

$$D_{g,i} = \frac{\sum_m \left|\frac{N_{g,m,i}}{N_{m,i}} - \frac{N_{g,i}}{N_i}\right| \cdot N_{m,i}}{2 \cdot N_i \cdot \frac{N_{g,i}}{N_i} \cdot (1 - \frac{N_{g,i}}{N_i})},$$

(9)

which is a normalized version of the mean deviance measure that takes into account the fact the different neighborhoods can have different population sizes.

The Gini Coefficient:

$$gini_{g,i} = \frac{\sum_m \sum_l |\frac{N_{g,m,i}}{N_{m,i}} - \frac{N_{g,l,i}}{N_{l,i}}| \cdot N_{m,i} \cdot N_{l,i}}{2 \cdot N_i^2 \cdot \frac{N_{g,i}}{N_i} \cdot (1 - \frac{N_{g,i}}{N_i})},$$

(10)

which can be interpreted as measuring the proportion of individuals of the other group experienced by group $g$. Unlike the mean deviance measure it is sensitive to redistribution among neighborhoods above or below the population mean demographics.

Finally, the exposure $B_{gg}$ index, also known as the correlation ratio (CR or $\eta^2$) or the mean squared deviation:

$$\eta_{g,i}^2 = \frac{\sum_m N_{g,m,i}^2}{N_{g,i} \cdot (1 - \frac{N_{g,i}}{N_i})} - \frac{\frac{N_{g,i}}{N_i}}{1 - \frac{N_{g,i}}{N_i}}.$$

(11)

This measure attempts to capture the probabilities of random members of each group interacting given the demographic distribution. It accounts for both neighborhood size and the movement of individuals between neighborhoods above and below the mean.

## Controlling for individual demographics

In order to control for individual demographics of IAT respondents, we transformed the individual bias responses into an indicator for $D_{biep} > 0$. This variable thus indicates whether the individual respondent had a positive bias for White faces or not. For each year, a logistic regression was performed that included the city-level variables of the natural logarithm of population, the majority groups size adjustment, and the segregation adjustment, and the individual level variables of race, educational attainment and birth sex. The 14-point educational attainment scale included with the IAT data, edu_14, was recoded into three categories of "High School Graduate or Below", "Some College or College Graduate", and "Advanced Degree". For some years there were no respondents in the "High School Graduate or Below" category, in which case that variable was excluded from analyzes. Self-reported racial demographics (raceomb before 2016 and raceomb_002 afterwards) was recoded to three categories of White, Black, and Multiracial, with other races and unknown combined as the base category.

## Comparison to previous results associating area deprivation with racial IAT responses

We downloaded the average maximum heat index (HI) in degrees Celsius for U.S. counties from the North America Land Data Assimilation System Daily Air Temperatures and Heat Index 1979-2011 database. This was the strongest predictor of between-city differences in implicit racial bias levels in a previously published analysis[53]. The maximum heat index was averaged across counties within each CBSA.

Those analyzes used a kitchen-sink approach with regularizing regressions to determine which variables were relevant to predicting these differences between cities. Since the variables identified there are indicative of levels of environmental, social, and economic disadvantage, we additionally evaluated the relevance of the Area Deprivation Index (ADI) to between-city differences in implicit racial

bias. The ADI summarizes neighborhood variation in socioeconomic indicators at small spatial units down to the census block level and includes factors related to income, educational attainment, employment, and housing quality[75]. We averaged nationally anchored ADI values at the county level across all counties in each CBSA.

In order to determine the effects of these measures of neighborhood disadvantage on implicit racial biases we conducted separate OLS regressions including city size, the diversity adjustment, the segregation adjustment, and the ADI or HI. Since ADI and HI data are not available for all CBSAs, we additionally conducted regressions without the ADI and HI included, but with the reduced sample size for which these data are available. We note that in those regressions with a reduced sample size, but without the inclusion of the ADI or HI the variance explained by city size, the diversity adjustment, the segregation adjustment are higher than in the full sample, and outperform previous analyzes which only include measures of neighborhood disadvantage[53].

### Estimates of the learning rate

Independent empirical estimates of the learning rate, $\alpha$, which governs the coupling between inter-group interactions and bias levels, were obtained directly from the two-step OLS regressions described in Equations (5) and (7). From Equation (5) we obtain an estimate of $\hat{\alpha}_{scaling} = \frac{\beta_1}{\delta}$. Confidence intervals for $\hat{\alpha}_{scaling}$ were obtained from the OLS confidence intervals for $\beta_1$. We note there may be other effects besides learning such as top-down hierarchical structures and variations in growth rates that may additionally contribute to differences in the empirical scaling exponent $\beta_1$ from the expected value of $\delta = \frac{1}{6}$. In addition, we obtain a second, independent estimate of the learning rate: $\hat{\alpha}_{diversity} = \beta_2$ based on Equation (3) of the main text.

Results from experimental interventions designed to simulate inter-group contact were used to further validate and bound these estimates of $\alpha$. We calculated the relative reduction in IAT $D_{biep}$ scores pre- and post-intervention for 18 different systematic interventions of various strength[25,76]. These interventions included having participants read stories of various lengths and vividness designed to affirm "White-bad" and "Black-good" associations, modifying the IAT to include additional "Black-good" and "White-bad blocks", simulating competition with White opponents and cooperation with Black teammates, having participants read about threatening scenarios and shown images of friends in those scenarios and reminding participants of prominent Black athletes positives contributions to society[25,26]. Importantly, all of these interventions occurred directly between IAT tests and are all positive in nature. In reality, inter-group interactions may not always be positive in nature, and they play out continuously at potentially irregular intervals relative to when a given individual makes a judgment or decision that is influenced by implicit racial biases. Consequently, these experimental interventions can be interpreted as an upper bound on the effects of one additional inter-group interaction when that interaction happens immediately before implicit bias levels are assessed.

### Granger causality analyzes

In order to evaluate evidence for temporal precedence between structural factors and implicit bias levels we employed Granger causality analyzes[59] as implemented in the python statsmodels library. These tests start by fitting a linear regression of one of the three variables of interest (population, diversity, and segregation) and implicit bias levels for a single city using 10 years of data. Next, another linear regression is fit with one variable lagged in time against the other variable. Evidence that changes in the lagged variable preceded changes in the other variable is evaluated based on an F-statistic calculated by the percent change in the squared residuals from the lagged model from the sum of squared residuals of the non-lagged model. This statistic is adjusted for the number of comparisons and the

degrees of freedom to obtain an F-statistic and p-value. We conducted this analysis across all 43 cities with 10 years of data and for each choice of which variable to lag. We repeated this for lags of 1, 2, and 3 years.

To summarize the results, we computed the percent of the 43 cities that show evidence ($p < .05$) for temporal precedence at each lag. Confidence intervals were computed by bootstrapping these percentages with replacement and computing the standard error. To combine evidence across the four segregation measures used, we averaged the percent for each measure and combined the standard errors according to:

$$\sigma_{combined} = \sqrt{\frac{\sum \sigma_i^2}{4}} \tag{12}$$

where $\sigma_i$ are the standard errors computed for each segregation measure.

### Reporting summary

Further information on research design is available in the Nature Portfolio Reporting Summary linked to this article.

### Data availability

All data needed to evaluate the conclusions in the paper are present in the paper, the Supplementary Materials, or are publicly available. Source data are provided with this paper. IAT data can be found at: https://doi.org/10.5334/jopd.ac. U.S. Census data can be found at: https://data.census.gov. CBSA delineation files can be found at: https://www.census.gov/programs-surveys/metro-micro/about/delineation-files.html. ADI data can be found at: https://www.neighborhoodatlas.medicine.wisc.edu/. North America Land Data Assimilation System Daily Air Temperatures and Heat Index data can be found at: https://wonder.cdc.gov/nasa-nldas.html. All data has been obtained according to the terms and conditions of the websites hosting them. Source data are provided with this paper.

### Code availability

All code needed to reproduce the analyzes included in this manuscript can be found at https://github.com/enlberman/implicit_biases_cities[77] (https://doi.org/10.5281/zenodo.10258104).

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

## Acknowledgements

This work is partially supported by NSF-2106013 that benefited A.J.S., S.S., and F.K. and S&CC-1952050 to M.G.B. S.S. was supported by the Austrian research agency (FFG) under project No. 873927 ESSENCSE. F.K. was partly supported by the EU Horizon Europe project MAMMOth (Grant Agreement 101070285). The authors thank Shige Oishi and Nicholas Epley for their helpful discussions during the preparation of the manuscript.

## Author contributions

A.J.S., L.M.A.B., and M.G.B. designed research; A.J.S. performed research; M.G.B. supervised research; and A.J.S., S.S., F.K., L.M.A.B., and M.G.B. wrote the paper.

## Competing interests
The authors declare no competing interests.
