## [Peer Review File · Nature Communications]

Evidence for Lower Implicit Racial Biases in Larger More Diverse and Less Segregated U.S. CitiesReviewers' Comments:

Reviewer #1:

Remarks to the Author:

Summary

This manuscript aims to tackle the challenging question of how the organization of cities systematically influences the implicit biases of individuals. The authors approach the problem from the perspective of urban scaling theory, using their existing model that relaxes the homogeneous mixing assumption of the standard scaling theory to allow individuals to belong to distinct groups whose mixing may exhibit homophilic or heterophobic tendencies. The authors examine how this extended scaling framework can be used to understand variations in inter-group interactions and the resultant implicit biases across cities. To do this, they construct a learning curve model for implicit bias using the expected number of inter-group interactions from their scaling model, which results in a relationship that predicts that (1) larger, (2) more diverse (smaller majority group size), and (3) less heterophobic cities have lower average levels of implicit biases. The relationships 1-3 are tested using data from the Implicit Association Test (IAT), which measures the difference in response times when subjects pair images of white and black faces with positive and negative words. The authors find that factors 1-3 are predictive of IAT responses even after controlling for race, birth-sex, and educational attainment. They also use their framework to estimate the learning rate associated with implicit biases, finding that two independent estimates of alpha are consistent and that the empirical value is consistent with previous experimental interventions.

Contribution and Impact

As the authors acknowledge, there is plenty of existing work on how social contexts can drive implicit biases, but there is relatively little work on how the structure of human organization can affect these biases through their influence on social contexts. The authors tackle the latter question, providing important results to address a research gap in the literature.

The scaling model central to the analysis in this paper has been used in the authors' recent work, which is clearly acknowledged by the authors. However, the learning curve analysis is new to this manuscript and represents a nice extension of the results one can derive using this formalism. Technically speaking this is very simple extension (raising the expected number of inter-group contacts to an exponent representing the learning rate), but it creates a nice bridge between purely structural aggregate measures of mixing and an individual-oriented psychological phenomenon.

I think this paper constitutes an interesting and novel contribution to the field of interdisciplinary urban science, which is my field of expertise. However, I can't speak for other fields such as sociology, psychology, neuroscience that may also be interested in this question. It appears to me that the authors did a comprehensive literature review covering related work in these other fields, but as these are not my areas of expertise I don't feel comfortable commenting on them.

(Potentially) Major concern: This being said, I don't see any significant theoretical advance taking place beyond incorporating the learning curve concept into the authors' existing heterogeneous mixing analyses, so I think it is critical that the empirical methodology is very solid and clearly confirms the predictions of the theory. I have a few concerns about this methodology, but if these concerns are dispelled by the authors then I think the work can have great interdisciplinary impact.

Exposition, clarity, and correctness

In general the paper is very clearly written and results are presented in an easily digestible manner.

The mathematical derivations and regression equations appear to be correct, and the learning curve extension of the model is consistent with previous learning curve models for similar phenomena.

Minor concern: The authors claim that "Despite the universality of implicit racial and ethnic biases in human societies and their well-documented detrimental effects, there have been no investigations of how the organization of people in cities may systematically influence them". I think this is a bit strong, given that substantial discussion in the paper is invested into comparing the results with Ref [46] which appears to have looked at a similar question (though, in my opinion, in a less elegant or principled way). I think maybe the authors should lighten this claim or justify why it's actually correct.

Minor concern: Can the authors further explain the concept of a noise ceiling in their context, and how it helps interpret the empirical results? There isn't much discussion of this outside of a very brief section in the Appendix, and it doesn't seem to be a very commonly used concept outside of neuroscience as far as I can tell.

Minor concern: The authors write "This suggests that these large-scale structural determinants of implicit racial biases are relevant for individuals' levels of bias. In other words, citywide organizational and structural characteristics influence individual implicit biases despite the diversity of local social environments that any individual urban inhabitant might encounter." I'm a bit confused by this. I thought that the citywide structure and organization would impact (with variation) the local social environments, and these local environments would influence individuals' implicit biases. Could the authors clarify what they mean here? It seems like the global \rightarrow local influence is implicit in the paper's empirical analyses, since global structural factors (city wide segregation indices and majority group sizes) are being used to assess the implicit biases of a typical individual in that city.

Methods and empirical evaluation

Despite the challenges merging large scale structural factors with individual behaviors, the IAT appears to be consistent with implicit biases the authors want to test. The regression analyses also appear to be methodologically solid in their execution.

Major concern: The authors remove the within-group interaction term in their analyses, claiming that the inter-group mixing term is more relevant. While this seems reasonable at first glance, it removes the h^{hom} dependence entirely, so there is no relative scale on which one can assess h^{het} . This therefore seems like it could end up being quite an important assumption. For example, what if structural factors not considered (e.g. poor infrastructure/urban design) are causing people to just interact less overall in a given city, regardless of their group identity. This causes h^{hom} and h^{het} to both decrease. But, in this case, by removing h^{hom} we only end up seeing a lower h^{het} , so think people simply prefer not to engage in inter-group mixing, when in fact people just don't like to engage in mixing at all. Could the authors elaborate on why this assumption is justified, and how it may impact their results?

Major concern: Following this logic, it seems that something like k_{inter}/k_{within} would be a more reasonable measure to assess the mixing diversity of an individual, since it accounts for both h^{hom} and h^{het} . Is there a reason why just using k_{inter} is a better measure that is more consistent with the literature on implicit biases? Is there a reason to assume that the number of within-group interactions or the relative number of within/inter-group interactions does not affect one's implicit biases?

Major concern: Another concern related to these, which I think is potentially even more important, is that the authors assume that h^{het} scales linearly with segregation indices. But the segregation indices listed are largely dependent on majority group size, which is supposedly an independent contribution in the regression analyses. Aren't the coefficients β_2 and β_3 then capturing

similar effects? I don't see in this case where mixing preferences come into play, or how the analysis is substantially different than if the authors just ran a correlation of implicit bias level versus segregation.

***Major concern*:** The authors mention how other sources of noise can affect the overall scaling exponent, which in turn affects the estimates of the learning rate. Can they elaborate on whether they think this noise is substantial or not when we consider estimating alpha in practice, perhaps with empirical estimates of this noise?

***Minor concern*:** I think the authors should include the usual disclaimer about the representativeness of the IAT samples. In this case, they may also want to compare the recorded individual metadata distributions (e.g. race) with the corresponding city-level distributions used in the analysis to demonstrate that they're similar.

*****Reproducibility***:**

The results appear to be reproducible given the details provided by the authors.

*****Overall Evaluation*****

I'd like to see a revised version of the manuscript where the authors either show that the concerns I have about the methodology and empirical evaluation do not impact the validity of the findings, or rework the relevant analyses to address these concerns. After these revisions, I think this study would be suitable to be considered for publication.

Reviewer #2:

Remarks to the Author:

This research establishes a link between between-city variation in implicit racial bias and the population, majority group size, and racial segregation of cities. This work is timely, connecting the structural with the social when both are at the forefront of current discourse. This work also appears to be rigorously conducted: I appreciate the authors' transparency, use of open data, and robustness checks across multiple operationalizations of constructs like residential segregation. Additionally, I think that there is a lot of value in model-based approaches like the authors used – which have, to date, not been applied much to study regional variation in psychological phenomena. All in all, there is a lot to like about this research, and it is positioned to make important contributions to several literatures.

My only issue with this manuscript, as written, is the authors' inappropriate use of causal language. Throughout the manuscript (abstract, introduction, discussion), and in the title itself, the authors interpret their findings to indicate that city population, majority group size, and residential segregation *drive* implicit biases. However, as far as I can tell, their analytic methods do not provide strong evidence for one causal pathway over the other (i.e., that implicit biases drive populations, segregation, etc). They ground their conclusion, in part, in previous research by Payne and colleagues, and by Hehman and colleagues. Payne and colleagues linked current variation in regional implicit bias with historical slavery conditions, and argued that slavery caused bias because the reverse relationship is temporally impossible. Hehman and colleagues selected environmental factors that are plausibly unlikely to be direct or indirect downstream consequences of bias, such as maximum heat index, thereby providing relatively strong evidence for environmental factors that cause bias rather than vice versa. In contrast, in the context of the present research, bias very likely contributes to total population, majority group size, and racial segregation. See Rentfrow et al. 2008 for an extensive discussion of the causal pathways that are possible here (e.g., selective migration, social influence). To

be sure, these are likely recursive relationships, such that bias influences segregation (for example), and segregation reinforces biases. And unless I am missing something (which I might be. I understand these specific equations but am not highly familiar with these methods), the present research does not provide strong evidence for the causal influence claimed by the authors. This issue should be easy enough to fix with small edits. As someone who is deeply invested in this area of research, I would like the authors' claim that "These effects suggest that as more people move into cities over the next decades, implicit biases will decrease, so long as cities do not become too segregated, remain centers of diversity, and residents continue to learn from shifting social environments." to be true as much as the authors would. However, the present research just can't support claims like this.

Reviewer #3:

Remarks to the Author:

This study investigates how structural features of cities (population size, segregation, and majority group size) correlate with implicit racial bias. It finds that cities with larger populations, less segregation, and more diversity have lower levels of implicit bias.

The aims of this paper are timely and relevant. Understanding which aspects of a context relate to implicit bias are important for advancing our theoretical understanding of why implicit biases persist despite declines in explicit bias and why they seem so resistant to change.

Overall, I'm enthusiastic about this paper. For transparency, I cannot comment on the analytical tools used or how they were implemented. The statistics were beyond my expertise and knowledge. However, I can comment on some of the theoretical and methodological aspects:

1. Past research has shown that structural features of the environment relate differently to the implicit biases of White versus Black respondents. For example, although segregation is associated with greater implicit bias among White respondents, it is associated with lower implicit bias among Black respondents. I know that the authors controlled for race in their analyses, but they do not examine interactions between race and structural features. In other words, the assumption of the model is that population size, segregation, and diversity have the same effect on the biases of Black and White respondents, but past findings suggest this may not be the case. This limits the interpretability of these findings.

2. In the introduction, implicit biases are defined as the differential treatment of individuals who belong to outgroups. However, implicit bias is a cognitive, not a behavioral, construct. Implicit biases link groups to evaluations or stereotypes, which is different from discrimination, or differential treatment based on group membership. In fact, implicit biases (at least at the individual level) very modestly predict discrimination.

3. The theoretical rationale for the analyses seemed underdeveloped. Is intergroup contact theorized to have the same effects for both Black and White residents? In addition, the introduction and methods do not speak to the nature of the contact. In some cases, the contact might be negative depending on other aspects of the environment (e.g., income inequality, status differences, competition for resources, etc.). Related to this point, I noticed that one of the measures of segregation used was the Gini coefficient, which is traditionally a measure of wealth or income inequality. I think that a brief explanation of each of the measures and why they were used would be helpful.

4. This comment is about the accessibility of the paper. Overall, there was a lot of jargon, which seemed unnecessary and potentially confusing. For example, "heterophobia" in the context of bias can sound like a fear of heterosexual people. Why not use common terms (like segregation) throughout?

5. It was unclear how the heat index was relevant in these analyses. I understand that past work found that to be a relevant factor predicting between-city differences in bias, but is there an explanation for why that is? How are the heat index and deprivation related?

6. Please be careful with the use of causal language (e.g., "driving subsequent implicit biases")

Our responses below are in **bold text**. Text drawn from the revised manuscript are in *italics*. Each point is separated by a horizontal line break. Changes and additions to the manuscript text are indented below our responses with page numbers indicated.

REVIEWER COMMENTS

Reviewer #1 (Remarks to the Author):

Summary

This manuscript aims to tackle the challenging question of how the organization of cities systematically influences the implicit biases of individuals. The authors approach the problem from the perspective of urban scaling theory, using their existing model that relaxes the homogeneous mixing assumption of the standard scaling theory to allow individuals to belong to distinct groups whose mixing may exhibit homophilic or heterophobic tendencies. The authors examine how this extended scaling framework can be used to understand variations in inter-group interactions and the resultant implicit biases across cities. To do this, they construct a learning curve model for implicit bias using the expected number of inter-group interactions from their scaling model, which results in a relationship that predicts that (1) larger, (2) more diverse (smaller majority group size), and (3) less heterophobic cities have lower average levels of implicit biases. The relationships 1-3 are tested using data from the Implicit Association Test (IAT), which measures the difference in response times when subjects pair images of white and black faces with positive and negative words. The authors find that factors 1-3 are predictive of IAT responses even after controlling for race, birth-sex, and educational attainment. They also use their framework to estimate the learning rate associated with implicit biases, finding that two independent estimates of alpha are consistent and that the empirical value is consistent with previous experimental interventions.

Contribution and Impact

As the authors acknowledge, there is plenty of existing work on how social contexts can drive implicit biases, but there is relatively little work on how the structure of human organization can affect these biases through their influence on social contexts. The authors tackle the latter question, providing important results to address a research gap in the literature.

The scaling model central to the analysis in this paper has been used in the authors' recent work, which is clearly acknowledged by the authors. However, the learning curve analysis is new to this manuscript and represents a nice extension of the results one can derive using this formalism. Technically speaking this is very simple extension (raising the expected number of inter-group contacts to an exponent representing the learning rate), but it creates a nice bridge between purely structural aggregate measures of mixing and an individual-oriented psychological phenomenon.

I think this paper constitutes an interesting and novel contribution to the field of interdisciplinary urban science, which is my field of expertise. However, I can't speak for other fields such as sociology, psychology, neuroscience that may also be interested in this question. It appears to me that the authors did a comprehensive literature review covering

related work in these other fields, but as these are not my areas of expertise I don't feel comfortable commenting on them.

(Potentially) Major concern: This being said, I don't see any significant theoretical advance taking place beyond incorporating the learning curve concept into the authors' existing heterogeneous mixing analyses, so I think it is critical that the empirical methodology is very solid and clearly confirms the predictions of the theory. I have a few concerns about this methodology, but if these concerns are dissipated by the authors then I think the work can have great interdisciplinary impact.

We appreciate that you recognize the potential impact of the manuscript and see its value and novelty. Addressing your comments has strengthened the paper as specified below. Thank you.

Exposition, clarity, and correctness

In general the paper is very clearly written and results are presented in an easily digestible manner. The mathematical derivations and regression equations appear to be correct, and the learning curve extension of the model is consistent with previous learning curve models for similar phenomena.

Minor concern: The authors claim that "Despite the universality of implicit racial and ethnic biases in human societies and their well-documented detrimental effects, there have been no investigations of how the organization of people in cities may systematically influence them". I think this is a bit strong, given that substantial discussion in the paper is invested into comparing the results with Ref [46] which appears to have looked at a similar question (though, in my opinion, in a less elegant or principled way). I think maybe the authors should lighten this claim or justify why it's actually correct.

We agree that Ref [46] is a less elegant and a less principled investigation albeit with similar goals. This sentence now reads (new text in bold):

*Despite the universality of implicit racial and ethnic biases in human societies and their well-documented detrimental effects, **existing studies lack a principled and theoretical basis to reveal** how the organization of people in cities may systematically influence **these biases.**"*

Minor concern: Can the authors further explain the concept of a noise ceiling in their context, and how it helps interpret the empirical results? There isn't much discussion of this outside of a very brief section in the Appendix, and it doesn't seem to be a very commonly used concept outside of neuroscience as far as I can tell.

Thank you for this suggestion. We have added the following to the main text and materials and methods:

On page 10:

To better understand the performance of our model, we employed estimates of the noise ceiling [48, 49]. Since implicit biases are inherently noisy attitudes [28], meaning that they fluctuate frequently, a model that is perfectly predictive may still fail to explain all of the observed variance and have an $R^2 < 1$. Noise ceiling estimates provide the maximum R^2 value that can be expected, given the level of noise in the data. Here, these estimates suggest that the three structural factors in our model may capture a majority of the variance than can be accounted for given the reliability [50] of the IAT measure (Supplementary Tables 25-36). As expected, because many U.S. cities are not so diverse, diversity accounts for more between-city variance in implicit biases than residential racial segregation (Figure 2B).

On page 15:

To better understand the performance of our model, we computed the bounds of the noise ceiling for the implicit bias measure. The idea of a noise ceiling is borrowed from cognitive neuroscience [49, 48] where the performance of predictive models can be limited by inherent noise in brain activity and measurement noise from human neuroimaging devices (e.g., functional magnetic resonance imaging or electroencephalogram). In those settings, a perfectly predictive model would only explain a fraction of the variance observed in the data (i.e., $R^2 < 1$). Therefore, without an estimate of the noise ceiling, it is impossible to assess whether a model fails to reach an R^2 close to 1 due to limitations in the model or the underlying measurement. This concept is not specific to brain imaging and can be applied to any measurement that is known to be noisy. Using noise ceiling estimates to evaluate models of implicit biases is appropriate because implicit biases are high entropy attitudes [60] and hence inherently more difficult to measure.

Minor concern: The authors write "This suggests that these large-scale structural determinants of implicit racial biases are relevant for individuals' levels of bias. In other words, citywide organizational and structural characteristics influence individual implicit biases despite the diversity of local social environments that any individual urban inhabitant might encounter." I'm a bit confused by this. I thought that the citywide structure and organization would impact (with variation) the local social environments, and these local environments would influence individuals' implicit biases. Could the authors clarify what they mean here? It seems like the global - local influence is implicit in the paper's empirical analyses, since global structural factors (city wide segregation indices and majority group sizes) are being used to assess the implicit biases of a typical individual in that city.

This is a great point. As you mention, the global and local scales are, in general, coupled. Regarding the first sentence, when conducting ecological level analyses, it is important to make sure that the mechanisms that are being tested are consistent from individuals up to entire cities. Thus, it is important that our empirical analyses, which include individual demographics, match our empirical analyses at the level of entire cities.

Regarding the second sentence, we are pointing out that the within-city variation around the expectation (e.g., is one neighborhood much more segregated than the citywide average) can be averaged while still explaining individuals' (mean) level of

bias. This does not mean that the local environment does not matter. However, as we demonstrate, one can account for much of the variance with global measures. Averaging over neighborhood variation within a city leads to highly predictive models across cities and within individuals.

Here is an example explaining our rationale in more detail. Consider a small city and a large city that have the same average levels of diversity and segregation. Let's take two example individuals from each city. Say that the small city person lives in a mixed neighborhood and the large city person lives in a very segregated neighborhood. Naively, we might think that the large city person living in a segregated neighborhood would have a higher implicit racial bias level than the small city person. However, if the large city is much larger in population than the small city, the person from the large city might have a similar, or possibly even lower, implicit racial bias level than the small city person living in a mixed neighborhood. This is due to the fact that individuals also move around cities, having experiences outside of their neighborhood of residence. Thus, the spatial structure of demographics and levels of diversity for the entire city are relevant to each individual. Again, this is not to say that the local environment doesn't matter, it says rather that the global environment also has a (strong) predictive influence, even when we ignore (or average over) individual neighborhood variation.

We have edited the sentence to read (new text in bold):

*This suggests that these large-scale structural determinants of implicit racial biases are relevant for individuals' levels of bias. In other words, citywide organizational and structural characteristics influence individual implicit biases despite the diversity of local social environments (e.g., **variation in neighborhood segregation compared to city-wide averages**) that any individual urban inhabitant might encounter.*

Methods and empirical evaluation

Despite the challenges merging large scale structural factors with individual behaviors, the IAT appears to be consistent with implicit biases the authors want to test. The regression analyses also appear to be methodologically solid in their execution.

Major concern: The authors remove the within-group interaction term in their analyses, claiming that the inter-group mixing term is more relevant. While this seems reasonable at first glance, it removes the h^{hom} dependence entirely, so there is no relative scale on which one can assess h^{het} . This therefore seems like it could end up being quite an important assumption.

The main theoretical thrust of our model is that inter-group interactions play a large and causal role in determining individuals' implicit racial bias levels. This is supported by a large body of existing psychological research that we cite throughout the manuscript (e.g., Ekstrom et al, 2022; Baron, 2006), where more interactions with out-group members leads to lowered implicit biases. In contrast, as far as we are aware, there is not much research to link within-group interactions to implicit bias levels. While of course there are many scenarios we can imagine in which within-group

interactions of a particular type might foster implicit biases (e.g., maybe interacting with only in-group individuals may increase out-group biases), there is no general proposed individual-level mechanism for such an effect.

Though there is some evidence that high levels of implicit biases are associated with increased homophilic tendencies (Jacoby-Senghor, 2015; Wimmer, 2010), these studies do not discuss alternative mechanisms for inducing changes in implicit biases other than changes in between-group interactions resulting from homophily.

We have added the following text on page 6 to make it more clear why we are only focusing on the between-group term in our model (new text in bold):

Though there is some evidence that high levels of implicit biases are associated with increased homophilic tendencies [33, 34], these studies do not discuss alternative mechanisms for inducing changes in implicit biases other than changes in inter-group interactions resulting from homophily. In contrast, there is a large body of previous research that has qualitatively demonstrated that inter-group interactions shape implicit racial biases [16, 24, 27, 28, 35, 36, 37, 38, 39]. Thus, we focus on this term to build our model.

In addition, we point out that h^{hom} and h^{het} both capture relative reductions (or increases) in interactions. In other words, a value of 0.2 for h^{het} means that interactions between groups are 20% less than what they would be otherwise. What they would be otherwise is what they would be given the diversity and population size of the city. It could be a 20% reduction from a large number or a small number. So yes, h^{het} and h^{hom} are relative and not absolute measures of mixing. It may be possible to design a model with absolute measures of mixing, but that likely requires a mathematical treatment of within-city human mobility flows that has not yet been developed (though we plan to pursue this line of research in the future).

For example, what if structural factors not considered (e.g. poor infrastructure/urban design) are causing people to just interact less overall in a given city, regardless of their group identity. This causes h^{hom} and h^{het} to both decrease. But, in this case, by removing h^{hom} we only end up seeing a lower h^{het} , so think people simply prefer not to engage in inter-group mixing, when in fact people just don't like to engage in mixing at all. Could the authors elaborate on why this assumption is justified, and how it may impact their results?

First, we note that higher values of h^{het} do not necessarily mean that people have strong preferences against inter-group mixing. Rather, they account for some combination of individual preferences and top-down structural factors (e.g., existing segregation that limits exposure to diverse individuals). In the present study, we focus on using residential racial segregation to proxy h^{het} , which is typically thought of as being driven more by top-down structural factors than individual preferences.

In this example, indirect effects of an overall reduction in mixing due to poor design would be captured by our model through residential segregation. However, the direct effects of poor urban design not be captured by our model unless larger cities

systematically have better or poorer urban design/infrastructure. This is because we specifically estimate h^{hom} and h^{het} as functions of residential racial segregation in our empirical model. Therefore, the effects of urban design would show up in the remaining variance that is not accounted for by our model (i.e., in the residuals). If we had measures of urban design and the quality of city infrastructure, they could be added to the model (and be included in the estimation of h^{hom} and h^{het}) and improve the predictions. However, at this point we cannot quantify their significance, if any. It is an interesting issue for future research.

***Major concern*:** Following this logic, it seems that something like k_{inter}/k_{within} would be a more reasonable measure to assess the mixing diversity of an individual, since it accounts for both h^{hom} and h^{het} . Is there a reason why just using k_{inter} is a better measure that is more consistent with the literature on implicit biases? Is there a reason to assume that the number of within-group interactions or the relative number of within/inter-group interactions does not affect one's implicit biases?

It is true that the ratio of k_{inter}/k_{within} may be a better measure of mixing diversity for an individual, however, this measure is not in-line with the exposure-based learning mechanism that we propose is at work in reducing implicit racial biases (e.g., in Lai, 2014, Lai 2016, Lai 2018). In expertise(exposure)-based learning, decreases in biases happen as people experience more interactions with out-group members and increases in biases happen as people have fewer interactions with out-group members. Thus, our model suggests that the quantity of k_{inter} alone is the important one to consider. We note that the quality of interactions is important as well, but we currently do not have good models to understand how city environments systematically impact interaction quality (nor do we have empirical data at scale that could be used to validate such a model). Importantly, the effects we observe, both here and in previous papers (e.g., Stier, 2021) suggest that on average these interactions are positive (both for in-group and out-group interactions) and lead to better outcomes (e.g., from depression, incomes, and implicit biases, etc.), but more work and data are needed.

To see how the literature supports this point of view we can play devil's advocate and assume that the ratio k_{inter}/k_{within} is the important quantity to set implicit bias levels. If that were true, then bringing people into the lab and exposing them to simulated out-group interactions (e.g., in Lai, 2014, Lai 2016, Lai 2018) would only have an effect on people for who k_{inter} was very close to k_{within} . When k_{inter} and k_{within} are not close, this ratio does not change much by changing k_{inter} .

Instead, these papers find a strong effect for most individuals after a single simulated exposure (albeit one that degrades quickly), meaning that it may not matter at all how large one's in-group social network is. Since the U.S. has 50%-50% racial demographics in few, if any, places, the strong effects observed in the lab are unlikely to be driven by the few people with a very balanced mix of between- and within-group social contacts.

To further demonstrate this point, we have included an explicit calculation of k_{inter}/k_{within} in the supplementary text on pages 3-5:

A Measure of Relative Mixing

Here, we compare our model in the main text to one which uses a measure of relative mixing. We start with the ratio:

$$k_{inter}/k_{within} \sim \frac{(\frac{N_1}{N} - (\frac{N_1}{N})^2)(2 - h_1^{het} - h_2^{het})}{(\frac{N_1}{N})^2(1 + h_1^{hom}) + (\frac{N_2}{N})^2(1 + h_2^{hom})}$$

If we let the homophily and heterophobia terms be zero, for the moment: where we have substituted $N_2 = N - N_1$. Expanding out the terms in the denominator and substituting $d = N_1/N - (N_1/N)^2$, with $d \in [0, .25]$, we can write:

$$k_{inter}/k_{within} \sim \frac{d}{1 - 2d}$$

We can directly compare this measure with the diversity part of k_{inter} , i.e. d by looking at the variance explained and information criteria of the two linear models, $\ln(bias) \in \ln(d/(1-2d))$ and $\ln(bias) \in \ln(d)$.

Supplementary Figure 1: The difference in R^2 (left) and AIC (right) for the two linear regressions $\ln(b) \sim \ln(k_{inter})$ and $\ln(b) \sim \ln(k_{inter}/k_{within})$. Distributions for these differences were computed via 500 bootstrap resamples with replacement. Horizontal lines show 95% bootstrap confidence intervals.

When comparing the R^2 and Akaike information criterion (AIC) across 500 bootstrap resamples, we find no significant advantage for either model (Supplementary Figure 1). The similarity between these two models is due to the fact that, in the absence of heterophobia or homophily, k_{inter}/k_{within} is a monotonic transformation of k_{inter} . In

other words, k_{inter}/k_{within} and k_{inter} necessarily are perfectly rank order correlated (i.e., $r_s = 1$).

In the presence of homophily and heterophobia the linear regression to predict bias becomes:

$$\ln(b) \sim \ln(d) + \ln(2 - h_1^{het} - h_2^{het}) - \ln\left(\left(\frac{N_1}{N}\right)^2(h_1^{hom} - h_2^{hom}) + \left(1 - 2\frac{N_1}{N}\right)(1 + h_2^{hom})\right) \quad (13)$$

When comparing this model to that used in the main text ($\ln(b) \sim \ln(d) + \ln(2 - h_{het}^1 - h_{het}^2)$, population term excluded) we again find no significant advantage for either model. However, there is a clear trend towards the model used in the main text, i.e., $\ln(b) \sim \ln(k_{inter})$ having higher adjusted R^2 and lower AIC (Supplementary Figure 2).

Supplementary Figure 2: The difference in R^2 (left) and AIC (right) for the two linear regressions $\ln(b) \sim \ln(k_{inter})$ and $\ln(b) \sim \ln(k_{inter}/k_{within})$ including heterophobia and homophily effects. Distributions for these differences were computed via 500 bootstrap resamples with replacement. Horizontal lines show 95% bootstrap confidence intervals.

Major concern: Another concern related to these, which I think is potentially even more important, is that the authors assume that h^{het} scales linearly with segregation indices.

This is an important point, thanks for raising it. Without additional data, e.g., cell phone mobility records tied to demographics alongside social contact tracking, we cannot directly measure h^{het} or test what the ‘true’ relationship between h^{het} and residential segregation should be. Below we conducted additional analyses where we model the relationship between h^{het} and segregation indices across various polynomials from degree 1 to 20. We find that the model with a 1-degree polynomial, i.e., the linear model used in the main text generally has the lowest (or close to the lowest) delta-AIC. This means that across all years and four choices of segregation measures (GINI, Mean Deviance, Exposure, and Segregation Index), the linear model was the least overfit. While the average delta-AIC increased as the degree of the model

increased, the distributions of delta-AIC were not significantly different from the linear model until a degree of 10.

On the other hand, we found that the adjusted coefficient of determination (R^2) was significantly larger for the quartic degree polynomials, though this difference was on the order of 5% (i.e., 8% to 13%) and no further benefit to R^2 was observed for polynomial degrees beyond 4.

These results suggest that, from a purely statistical perspective, a **4th degree polynomial is the best choice** with respect to minimizing overfitting, maximizing variance explained, and having the fewest parameters. Importantly, even for the model with the 4th degree polynomial, the linear component of that model was still important/predictive (i.e., the polynomial term did not soak up all of the variance).

Note that this choice of 4th degree polynomial to link segregation measures and h^{het} improves the estimated variance explained by the full model with all three effects. The average now is 40% which is 72% of the noise ceiling median lower bound. Compare this to 35% and 63% of the noise ceiling median lower bound for a linear coupling (degree 1) between segregation and h^{het} .

As we mention above, we have no theoretical basis to justify using a 4th degree polynomial to model the relationship between segregation measures and h^{het} . Thus, we stick to results with the linear model in the main text, but have now added additional information and analyses using these higher order polynomials to map the relationship between h^{het} and residential segregation

We made the follow additions to the text:

On page 8 (new text in bold):

*We measured heterophobia values, h^{het} , as linearly dependent (see **Supplementary Figure 2 for equivalent analyses utilizing non-linear dependences**) on residential racial segregation calculated from racial demographics in census tracts (small areas of 4,000 inhabitants).*

On page 6 of the supplement:

Continued on next page

Figure 5: Comparison of higher order coupling between h^{het} and segregation indices. Statistically, a 4th degree polynomial is the best choice with respect to minimizing overfitting, maximizing variance explained, and having the fewest parameters. A 4th degree polynomial improves the estimated variance explained by the full model with all three effects to an average of 40% (72% of the noise ceiling median lower bound) compared to 35% (63% of the noise ceiling median lower bound) for a linear coupling (degree 1). a) That the model with a 1-degree polynomial, i.e. the linear mode used in the main text has the lowest average delta-AIC. This means that across all years and choices of segregation measure, the linear model is the least overfit model or close to as good as the least overfit model (i.e., the linear model AIC is lowest or slightly larger than the lowest AIC). The distributions of delta-AIC are not significantly different from the linear model until a degree of 10 (solid v.s. transparent). b) The adjusted coefficient of determination (R^2) is significantly larger (transparent v.s. solid) for quartic degree and higher polynomials. This difference is on the order of 5% and no further benefit to R^2 is observed for degrees beyond 4.

But the segregation indices listed are largely dependent on majority group size, which is supposedly an independent contribution in the regression analyses. Aren't the coefficients β_2 and β_3 then capturing similar effects?

It is true that the majority group size and segregation values are correlated, but only for some of the segregation measures. For the mean deviance and exposure measures, majority group size and segregation are negatively correlated (-.84 and -.66 respectively). However, the segregation index and Gini index measures are not significantly correlated with majority group size ($r = -0.03, p = 0.31$ and $r = -0.029, p = 0.39$, respectively).

Nonetheless, our model uses non-linear transformations of these variables which reduces their co-linearity, and allows us to estimate their independent effects. This was part of our intention by selecting multiple segregation indices, some of which are more and some of which are less correlated with majority group size. Our statistical methodology accounts for the independent effects of segregation while controlling for

the majority group size, and vice versa. We can measure the degree to which the correlation between these two measures contaminates our beta_1 and beta_2 estimates via the variance inflation factor (VIF). We find that except for the Mean Deviance segregation measure, all VIF values are below 5. This indicates that the collinearity between the majority group size measure and the segregation measures is not so high as to prevent the regression from estimating separate, independent effects. In other words, we can statistically separate the effects of majority group size and segregation.

We added the following text:

On pages 8-9 (new text in bold):

*In addition, more diversity and higher levels of residential racial segregation are significantly related to scaling deviations (Supplementary Table 1) and associated with higher average IAT scores, in line with Equation 3. **Importantly, the diversity and segregation terms can be statistically separated even though they are correlated for some measures of segregation (Supplementary Figure 3).***

On page 7 of the supplement:

Continued on next page

Supplementary Figure 3: Variance Inflation Factors (VIF) between the majority group size and segregation term of the model are low. Across all segregation measures, with the exception of the mean deviance measure, the estimated VIF is below 5. This indicates that colinearity between the majority group size measure and the segregation measures is not high enough to prevent the regression model from estimating separate, independent effects. In other words, majority group size and segregation effects are statistically separable even though they are correlated for some pairs of measures.

I don't see in this case where mixing preferences come into play, or how the analysis is substantially different than if the authors just ran a correlation of implicit bias level versus segregation.

Just to reiterate, higher values of h^{het} do not necessarily mean that people have strong preferences against out-group mixing. Rather, they signal some combination of individual preferences and top-down structural factors that result in overall reduced mixing. This is very important to consider, as structural factors do matter for real life agents with limited knowledge and experience.

With the available data, we cannot directly observe levels of mixing between groups. From past research we know that segregation tends to reduce mixing. This is also supported by most existing theories of residential segregation. Thus, we use it as a

proxy for reduced mixing in our model. In the future, data may also be available that will help us to test more complex models (i.e., cellphone trace data with user demographics) and to empirically measure between-group mixing.

You are correct that our interpretation of the direction and significance of the relationship between segregation and bias would be similar with a (Spearman) correlation. Our statistical model is log-linear, which is a monotonic transformation of the variables.

By grounding this model in theory, we can interpret the coefficients of the model. In particular, the coefficient on city population tells us about a mixture of social network density and learning effects, the coefficient on the diversity term tells us about learning effects, and the coefficient on the segregation term tells us about a mixture of learning effects and how segregation and group mixing are coupled. In other words, if we had only reported bivariate spearman correlations between these three factors and bias levels, we could only make qualitative claims about their relationships and the mechanisms driving those relationships. Instead, our model provides precise quantitative interpretations of the nature of these relationships. Arguably more important, our model estimate of the learning rate α , provides a precise quantitative prediction about the psychological mechanism underlying implicit biases that can be expanded on in future studies.

We have added the following text on page 8:

The choice to proxy heterophobia with segregation measures is motivated by past empirical [46, 47] and theoretical work (e.g., [48, 49]) linking population mixing and segregation.

Major concern: The authors mention how other sources of noise can affect the overall scaling exponent, which in turn affects the estimates of the learning rate. Can they elaborate on whether they think this noise is substantial or not when we consider estimating α in practice, perhaps with empirical estimates of this noise?

To be specific, in the manuscript we stated: *“In addition, we note that there may be other sources of deviations from the expected scaling exponent of $\delta = 1/6$ including top-down hierarchical constraints on inter-group interactions (42), growth rate fluctuations, and other higher-order effects (43), which may contribute to differences in independent estimates of α calculated from the first and the second terms of Equation 3.”*

The intent of this statement was to convey the fact that there may be some systematic or organizational characteristics of cities that may impact observed scaling exponents (e.g., in reference 42 the impact of hierarchical authority structures is discussed in the context of medieval cities with different ranks of inhabitants who all reported up to kings or land-owners). However, one nice property of our model is that we have two estimates of the learning rate, one of which is independent of these other potential sources of deviations (i.e., the majority group size term). Specifically, the estimate of the learning rate from the majority group size term is not dependent on any of these

factors, because absolute population size does not explicitly enter this term. In contrast, the estimate of the learning rate from the city population size term may be more imprecise since we do not know which unmodelled factors that impact the scaling exponent may be at play. Specifically, our model says that the coefficient on the population term is equal to $\delta \cdot \alpha$. We assumed $\delta = 1/6$ to calculate this estimate of α . However, we know that hierarchical social structures and differing growth rates between urban characteristics (e.g., population and wages) can lead to deviations in δ away from $1/6$. Importantly, we find convergence of this estimate of the learning rate, α , with the majority group size term estimate of α , which suggests that any such deviation of δ away from $1/6$ is small. In other words, the implied variation around $\delta = 1/6$ is small.

Minor concern: I think the authors should include the usual disclaimer about the representativeness of the IAT samples. In this case, they may also want to compare the recorded individual metadata distributions (e.g. race) with the corresponding city-level distributions used in the analysis to demonstrate that they're similar.

Thank you for this suggestion, we have added the following text:

On page 8 (new text in bold):

We note that CBSAs are functional definitions that capture the spatiotemporally extended social networks of cities and include, in the same unit, where people live, socialize, and work [45].

In addition, it is important to note that these IAT data are not nationally representative and tend to be younger, more educated, and with a higher percentage of female participants. This likely leads to underestimates of bias levels [46]. Nonetheless, racial demographics are strongly correlated across cities (Supplementary Table 1 & Supplementary Figure 1) suggesting that this sample is suitable for relative comparisons across cities.

On page 5 of the supplement:

Continued on next page

Supplementary Figures

Supplementary Figure 1: Census and IAT Demographics and strongly correlated. Data for 2020 demonstrates the strong correlation between the fraction of city populations that indicated their race as white or black in the U.S. census data and the iat sample data. Note that while the x and y axis scales are similar for the fraction of the population identified as white, the IAT sample has fewer black respondents than would be expected from a representative sample. Nonetheless, the strong correlations indicate that the IAT sample is suitable for relative, between city analyses.

On page 9 of the supplement:

Supplementary Table 1: Spearman rank order correlations between empirical population fractions and IAT sample fractions across cities. Correlations across cities between e.g., the fraction of the population with race reported as black in the U.S. census data and the fraction of the city's IAT sample which self reported their race as black. While the IAT sample is not nationally representative, these correlations suggest that the IAT sample does capture relative demographic differences between cities with respect to race.

year	r_s (white)	p value	r_s (black)	p value	# cbsas
2010	0.91	2.03e-25	0.93	2.93e-29	64
2011	0.91	5.58e-27	0.94	7.21e-32	67
2012	0.92	2.12e-27	0.94	5.86e-32	67
2013	0.93	1.13e-29	0.93	4.73e-31	68
2014	0.90	5.95e-28	0.93	4.01e-33	76
2015	0.90	1.45e-31	0.94	1.66e-39	85
2016	0.83	1.85e-24	0.93	1.04e-39	91
2017	0.89	2.51e-37	0.95	8.36e-55	107
2018	0.90	9.83e-41	0.96	1.05e-60	111
2019	0.90	1.08e-43	0.96	4.95e-66	119
2020	0.92	7.11e-51	0.96	1.39e-65	119

Reproducibility:

The results appear to be reproducible given the details provided by the authors.

Overall Evaluation

I'd like to see a revised version of the manuscript where the authors either show that the concerns I have about the methodology and empirical evaluation do not impact the validity of the findings, or rework the relevant analyses to address these concerns. After these revisions, I think this study would be suitable to be considered for publication.

Thank you for your constructive comments. Addressing them has greatly improved the manuscript.

Reviewer #2 (Remarks to the Author):

This research establishes a link between between-city variation in implicit racial bias and the population, majority group size, and racial segregation of cities. This work is timely, connecting the structural with the social when both are at the forefront of current discourse. This work also appears to be rigorously conducted: I appreciate the authors' transparency, use of open data, and robustness checks across multiple operationalizations of constructs like residential segregation. Additionally, I think that there is a lot of value in model-based approaches like the authors used – which have, to date, not been applied much to study regional variation in psychological phenomena. All in all, there is a lot to like about this research, and it is positioned to make important contributions to several literatures.

My only issue with this manuscript, as written, is the authors' inappropriate use of causal language. Throughout the manuscript (abstract, introduction, discussion), and in the title itself, the authors interpret their findings to indicate that city population, majority group size, and residential segregation *drive* implicit biases. However, as far as I can tell, their analytic methods do not provide strong evidence for one causal pathway over the other (i.e., that implicit biases drive populations, segregation, etc).

Thank you for your positive feedback. We agree that our use of causal language throughout the manuscript was too strong and thank you for bringing this point up. It has led us to reframe portions of the manuscript. We believe that the structural factors that we model are likely co-determined with implicit bias levels and that multiple bi-directional causal pathways exist. To address this, we have added an additional section to the manuscript to uncover the timescales of temporal precedence between our structural factors of interest and implicit bias levels.

In short, we find much stronger evidence that changes in structural factors precede changes in implicit biases year-to-year. However, once we zoom out to a three-year time-lag, there is equal evidence for both causal directions (i.e., population/segregation/diversity -> bias, and bias -> population/segregation/diversity). In many ways, this result makes sense: given the rapid learning involved in setting

implicit biases, we expect people to psychologically adjust to changing social conditions faster than they can move to different neighborhoods or cities. Given these analyses, we have qualified causal language to better match the available evidence. We reproduce these analyses below in more detail.

We have made the following changes:

The title now reads:

City Population, Majority Group Size, Residential Segregation, and Implicit Racial Biases in U.S. Cities

The abstract now reads (new text in bold):

*Implicit biases - differential attitudes towards out-group members - are pervasive in human societies. These biases are often racial in nature and create inequities across many aspects of life. Recent research has revealed that implicit biases are, generally, driven by social contexts. However, it is unclear if the regular ways that humans self-organize in cities systematically influence implicit racial bias strength. We leverage extensions of the models of urban scaling theory to predict and test between-city differences in these biases. Our model links spatial scales from city-wide infrastructure to individual psychology to predict that cities that are more populous, more diverse, and less segregated are less biased. We find broad empirical support for these predictions in U.S. cities with data spanning a decade from millions of individuals. **In addition, we find evidence that changes in cities social environments generally precede changes in implicit biases at short time-scales, but this relationship is bi-directional at longer time-scales.** We conclude that the **social** organization of cities strongly **influences** the strength of these biases and provides potential systematic intervention targets for planning more equitable societies.*

They ground their conclusion, in part, in previous research by Payne and colleagues, and by Hehman and colleagues. Payne and colleagues linked current variation in regional implicit racial bias with historical slavery conditions, and argued that slavery caused bias because the reverse relationship is temporally impossible. Hehman and colleagues selected environmental factors that are plausibly unlikely to be direct or indirect downstream consequences of bias, such as maximum heat index, thereby providing relatively strong evidence for environmental factors that cause bias rather than vice versa.

We note that in addition to the work by Payne and colleagues, we draw on additional observational evidence associating social contexts with bias levels and experimental work in which simulated changes in social environments caused (temporary) changes in bias levels (Citations in the manuscript numbered 16, 24, 27, 28, 33-37). Thus, our model and conclusions are based on a large body of literature from a diversity of sources of evidence (both causal and correlational) linking social context changes with implicit bias level changes.

In this manuscript, we have tried to refrain from analyzing potential environmental factors that are not in our explicit mathematical model. By doing so we restrict our analysis to factors that fit into the larger urban scaling literature which has ample evidence to justify the mechanistic claims made by urban scaling mathematical models. We cite Hehman et. al. to (1) address previous related work, and (2) to demonstrate how data driven analyses without explicit theory to back them up can fail to capture underlying mechanisms. To quote from pages 9 and 10 of the manuscript (new text in bold):

*Along these lines, other research has identified environmental variables related to area deprivation associated with inter-city variance in implicit racial bias [51]. However, with our model, we find that measures of area deprivation independently explain only a small portion of the variance in inter-city differences above and beyond the three structural factors we identify here (Supplementary Tables 15-18). This suggests that the area deprivation variables identified previously actually capture a combination of city population, segregation, and diversity (e.g., see **Supplementary Figure 5**) and that there are other factors, for example, segregated mixing in ambient populations [52], that may explain the remaining inter-city variance in implicit biases.*

For example, in the new Supplementary Figure 5 (see below) we demonstrate that the Maximum Heat Index suggested by Hehman, et. al. is strongly correlated with the percentage of black residents in cities. Combined with the analyses in Supplementary Tables 15-18, this suggests that the correlation between Maximum Heat Index and implicit bias levels is driven by the relationship between heat and diversity (and not simply heat alone), which may, in turn, be due to historical differences from slavery, crops grown (e.g., cotton), and politics between northern and southern states.

We added the following figure to page 9 of the supplement:

Continued on next page

Supplementary Figure 5: Max Heat Index and racial demographics are correlated. Using data from 2019 we demonstrate that the Average Daily Maximum Heat Index is significantly correlated to the percent of city populations that are Black. Combined with the analyses in Supplementary Tables 15-18, this suggests that the correlation between heat and implicit bias levels is driven by the relationship between heat and diversity, which may, in turn, be due to the historical difference in slavery, crops grown (e.g., cotton), and politics between northern and southern states.

In contrast, in the context of the present research, bias very likely contributes to total population, majority group size, and racial segregation. See Rentfrow et al. 2008 for an extensive discussion of the causal pathways that are possible here (e.g., selective migration, social influence). To be sure, these are likely recursive relationships, such that bias influences segregation (for example), and segregation reinforces biases.

And unless I am missing something (which I might be. I understand these specific equations but am not highly familiar with these methods), the present research does not provide strong evidence for the causal influence claimed by the authors.

We agree that there are bi-directional causal pathways likely at work here.

In fact, our theoretical model has a particular logic that does imply a specific causal pathway. The urban scaling model which gives numbers of interactions between groups is explicitly statistical and deals in averages over time. In that part of the model, the growth of cities spatially, socially, and in populations are concomitant and urban scaling relations are self-consistent, resulting from the structure of costs and benefits for urban agents in interaction over urban built-up spaces (Bettencourt, 2020). No result on the scales of cities and data observation is purely one-directional.

All of this is to say that the urban scaling portion of the model does not isolate a specific causal direction between city population and the number of between-group interactions; it provides an equilibrium feature of cities in terms of expected social interaction rates, for example. When we link these quantities to implicit biases, we do so through a model in which individuals adapt their psychology via learning from interactions. Here the causal pathway is explicit: more interactions → change in bias levels. We acknowledge that there are additional pathways mentioned by the reviewer such as selective migration and historical white flight that provide mechanisms for the inverse causal path. These pathways are not explicitly part of our model and arguably happen over relatively longer periods of time. We now acknowledge them throughout the manuscript and in our new temporal precedence analyses.

In the new section to analyze temporal precedence (“Granger causality”) we examine whether there is more evidence for changes in structural factors preceding changes in implicit biases or vice versa. These analyses have revealed that at a lag of 1 year there is significantly more evidence for changes in structural factors preceding changes in implicit bias (see Figure 4 and Table 1 below). However, at a lag of 3 years there is equal evidence for both directions, i.e., structural factors preceding changes in bias and changes in bias preceding changes in structural factors such as diversity and overall population size.

We made the following changes to the text:

One pages 12 to 15 we have added an additional section:

Timescales of Temporal Precedence

The learning mechanism linking biases and between-group interactions emphasizes a specific causal direction in the model: interactions → bias levels. However, there are other mechanisms, such as selective migration [56] and individual mixing preferences, that may facilitate reverse causal pathways in which bias levels influence changes in diversity, population size, and segregation, respectively. While the heterophobia term in the model can account for the effects of mixing preferences (along with historical processes and explicit racism, e.g., that influenced unfair lending policies), our model does not explicitly account for processes in which implicit biases facilitate changes in city diversity and population size.

To begin to understand the role of each of these causal directions, we take advantage of the fact that 43 cities have implicit racial bias data available for all 10 years of data. We employ Granger causality [57] to statistically test whether changes in one variable precede or follow changes in another variable. In brief, these analyses test whether the linear regressions between two variables of interest improve when one of the variables is lagged in time (see Materials and Methods). We perform these analyses for each city and calculate the percentage of cities with statistically significant evidence of temporal precedence.

We find evidence that changes in population size, diversity, and segregation precede changes in implicit biases at a lag of one year for a majority of cities (Table 1, Figure 4). In contrast, only a fraction of cities show evidence for the reverse temporal precedence. Results are similar at a lag of 2 years. At a lag of 3 years, however, there is equal evidence for both temporal precedence directions.

In combination with the mathematical model presented here, these results suggest a mismatch in the timescales at which different mechanisms play out. In particular, these analyses suggest that, at short timescales (i.e., 1 - 2 years), changes in structural factors primarily precede changes in implicit racial biases as individuals learn from and internalize changing social contexts. However, there is also some evidence of the reverse temporal direction at these short timescales. This direction of bias changes preceding structural factors may be due to immediate, individual-level effects such as changes in bias levels leading to changes to individual mixing preferences.

At long timescales, evidence is present for influence in both directions from biases to structural factors and vice versa. This fits with our model's suggestion of rapid learning involved in setting implicit biases (Figure 3). Psychological adaptations to changing social conditions are expected to be faster than the speed with which individuals (and their households) can move to different neighborhoods or cities. Thus, we expect that changes in biases happen faster than changes in city demographics and patterns of segregation. More work is needed to enumerate potential mechanisms linking bias levels back to structural changes (e.g., selective migration [56]) and to mathematically model these mechanisms in an urban scaling context.

Continued on next page

Figure 4: Granger causality analyses provide differing amounts of evidence for each direction temporal precedence across 43 cities. At lags of one and two years, more cities have evidence of changes in population preceding changes in bias. At a lag of three years, there is equal evidence for both directions. Error bars represent the bootstrapped standard error. The inset shows the same measure for diversity (dotted line) and segregation (dashed line) with Granger causality directions indicated by the same colors.

	1 year lag	2 year lag	3 year lag
population→bias	73.8 ± 7.0	78.6 ± 6.3	85.7 ± 5.5
bias→population	19.0 ± 5.9	35.7 ± 7.5	76.2 ± 6.2
diversity→bias	61.9 ± 7.3	66.7 ± 7.3	88.7 ± 4.9
bias→diversity	24.4 ± 6.7	45.2 ± 7.7	84.5 ± 5.6
segregation→bias	69.0 ± 7.1	76.2 ± 6.7	95.2 ± 3.3
bias→segregation	19.0 ± 6.3	38.1 ± 7.5	81.0 ± 5.9

In the discussion on page 15 (new text in bold):

The model developed here demonstrates that relatively simple considerations of heterogeneous mixing among a small number of social groups can explain a large proportion of why people in some cities have stronger implicit racial biases than in others. While it is somewhat surprising that only three factors - city population, majority group size, and racial segregation – account for so much between-city variance in bias levels, this is in line with recent evidence that implicit racial biases are driven more by social contexts than by individual differences in attitudes [58, 59]. Importantly, our

model and empirical evidence suggest that, at short timescales, implicit racial biases emerge from the interaction between city-wide social contexts that are shaped by the built environment and individual psychology, which determines how much and how quickly people learn from those contexts. At longer timescales, we find evidence that other, still unenumerated, and unmodeled mechanisms create feedback loops in which implicit racial biases shape these social contexts, e.g., through selective migration [56].

In the methods on page 22-23:

In order to evaluate evidence for temporal precedence between structural factors and implicit bias levels we employed Granger causality analyses [57] as implemented in the python statsmodels library. These tests start by fitting a linear regression of one of the three variables of interest (population, diversity, and segregation) and implicit bias levels for a single city using 10 years of data. Next, another linear regression is fit with one variable lagged in time against the other variable. Evidence that changes in the lagged variable preceded changes in the other variable is evaluated based on an F-statistic calculated by the percent change in the sum of squared residuals from the lagged model from the sum of squared residuals of the non-lagged model. This statistic is adjusted for the number of comparisons and the degrees of freedom to obtain an F-statistic and p-value. We conducted this analysis across all 43 cities with 10 years of data and for each choice of which variable to lag. We repeated this for lags of 1, 2, and 3 years.

To summarize the results, we computed the percentage of the 43 cities that show evidence ($p < .05$) for temporal precedence at each lag. Confidence intervals were computed by bootstrapping these percentages with replacement and computing the standard error. To combine evidence across the four segregation measures used, we averaged the percent for each measure and combined the standard errors according to:

$$\sigma_{combined} = \sqrt{\frac{\sum \sigma_i^2}{4}} \quad (12)$$

where σ_i are the standard errors computed for each segregation measure.

This issue should be easy enough to fix with small edits. As someone who is deeply invested in this area of research, I would like the authors' claim that "These effects suggest that as more people move into cities over the next decades, implicit biases will decrease, so long as cities do not become too segregated, remain centers of diversity, and residents continue to learn from shifting social environments." to be true as much as the authors would. However, the present research just can't support claims like this.

We have modified that sentence to read as follows. Given the additional analyses added above, we feel that this statement stands up to the empirical results of the paper.

These results suggest that as more people move into cities over the next decades, cities must not become too segregated, remain centers of diversity, and residents must continue to learn from shifting social environments if implicit biases are to decrease.

Thank you for your comments and suggestions. Addressing them has helped to greatly improve the manuscript.

Reviewer #3 (Remarks to the Author):

This study investigates how structural features of cities (population size, segregation, and majority group size) correlate with implicit racial bias. It finds that cities with larger populations, less segregation, and more diversity have lower levels of implicit bias.

The aims of this paper are timely and relevant. Understanding which aspects of a context relate to implicit bias are important for advancing our theoretical understanding of why implicit biases persist despite declines in explicit bias and why they seem so resistant to change.

Overall, I'm enthusiastic about this paper.

Thank you for your enthusiasm and for seeing value in our work.

For transparency, I cannot comment on the analytical tools used or how they were implemented. The statistics were beyond my expertise and knowledge. However, I can comment on some of the theoretical and methodological aspects:

1. Past research has shown that structural features of the environment relate differently to the implicit biases of White versus Black respondents. For example, although segregation is associated with greater implicit bias among White respondents, it is associated with lower implicit bias among Black respondents. I know that the authors controlled for race in their analyses, but they do not examine interactions between race and structural features. In other words, the assumption of the model is that population size, segregation, and diversity have the same effect on the biases of Black and White respondents, but past findings suggest this may not be the case. This limits the interpretability of these findings.

Thank you for this comment. We were not sure which papers you were referring to, so please let us know if the following reply does not address the findings that you had in mind.

One example that we cite in the manuscript comes from Payne et. al., 2019 in which current bias levels were associated with the local proportion of slaves in the population in 1860. Here they found that at the state and county level white respondents in areas with higher historical slave populations had higher pro-white biases but that black respondents in the same areas had higher pro-black biases.

While at first glance this would seem to be a differential effect for black versus white respondents, it can be fully explained by our model. Specifically, the researchers also reported that slave populations in 1860 were positively correlated with current segregation levels and with the current diversity of local areas. Under our model, we predict that areas with higher historical slave populations would have lower levels of between-group interactions, which would result in higher levels of out-group biases across for both black and white respondents. Since more positive IAT scores represent greater out-group biases for white respondents and more negative IAT scores represent greater out-group biases for black respondents, our predictions match the results of Payne et. al., 2019. In other words, we find the same directional results in our individual level models that include participants' race and ethnicity as predictors. Nonetheless, the structural factors of interest show the same direction across individuals even while black participants tend to have more negative bias scores and white participants tend to have more positive bias scores.

We agree that future work should strive to break these effects down across local areas and different groups. Our model is simpler and deals only in averages over the entire city, but it is impressive that even with this simplicity, the model is still highly predictive. We are actively working to extend the model by developing new mathematics based on fluid mechanics to allow for modeling sub-groups and neighborhoods of cities separately (i.e., not treating each city as a monolith).

2. In the introduction, implicit biases are defined as the differential treatment of individuals who belong to outgroups. However, implicit bias is a cognitive, not a behavioral, construct. Implicit biases link groups to evaluations or stereotypes, which is different from discrimination, or differential treatment based on group membership. In fact, implicit biases (at least at the individual level) very modestly predict discrimination.

Thank you for this clarification. We have updated the introduction with the following text (new text in bold):

*Implicit biases refer to **differential attitudes towards** individuals who belong to outgroups, in ways that are automatic.*

3. The theoretical rationale for the analyses seemed underdeveloped. Is intergroup contact theorized to have the same effects for both Black and White residents?

There is ample evidence that intergroup contact is a strong determinant of implicit bias levels (e.g., please see citations in the manuscript numbered 16, 24, 27, 28, 33-37). Notably, this evidence spans racial and ethnic groups. For our model, we start from the empirically backed assumption that implicit biases are generated via a learning process (e.g., please see citations in the manuscript 17-20). In other words, we assume that implicit biases are generated via a single mechanism that does not differ across groups of people.

That being said, this one mechanism, expertise-based learning, does not address the quality of contact (as we discuss below), and does not address individual differences that may contribute to different outcomes from “similar” interactions. Specifically, our model asserts that when an individual has a positive out-group interaction bias levels decrease with a certain power-law learning rate. Similarly, when an individual has a negative out-group interaction bias levels increase.

In reality, existing bias levels and the path dependence (your lifetime of outgroup experiences matter) implied by our model suggest that two individuals may not agree on the quality of an interaction. To this point, previous studies have found that implicit biases predict readiness to perceive threat in others’ facial expressions (Hugenberg, 2003)

While these considerations are important, we are not able to test them with the data that we have currently. In addition, we do not currently have the mathematics developed to describe these issues of interaction quality, though we hope to work on this over the coming years.

We have added a description of these limitations in the discussion on page 15 and 16:

An important implication of the results presented here is that, on average, inter-group contact in cities is beneficial with respect to reducing implicit racial biases. This matches results from the urban scaling literature that includes psychological depression [62], economic outputs [63, 1, 2], and creative outputs [64]. In all of these cases, the observation of increasing beneficial returns to city population suggests that interactions across these modalities are, on average, positive. However, the equations of urban scaling theory as formulated do not address interaction quality directly. Despite this, there is likely great variation in interaction quality within cities. For example, inter-group contact may be cognitively costly [65], and interactions between individuals or in certain neighborhoods may be negative, particularly in areas with high levels of existing implicit racial biases [66]. Thus, investigations of whether and how cities systematically facilitate interactions of differing quality are natural next steps.

In addition, the introduction and methods do not speak to the nature of the contact. In some cases, the contact might be negative depending on other aspects of the environment (e.g., income inequality, status differences, competition for resources, etc.).

This is a good point, thank you. This is a limitation of our model and also of the entire urban scaling literature which does not address systematic variation in the quality of interactions between cities or between places within cities, except for the statistics of residuals, which must be analyzed in detail. If such effects produce city size biases, they would be accounted for by size dependent variations (Bettencourt 2020), which has a “fluctuations-driven” effect on scaling exponents. Nevertheless, the success of the quantity-based model (which does not account for quality), which we present here suggests that for entire cities, on average, these intergroup interactions are positive. This is also true for the other urban scaling literature on GDP, wealth, and depression where more interactions all seem to lead to “better” outcomes (Bettencourt, 2013;

Stier, 2021). This suggests that, on average, these increased social interactions tend to be positive in nature (otherwise underlying social structures would dissolve). Thus, increases in wealth, GDP, innovation and decreases in depression are self-consistently associated with desirable sociality in terms of greater number of (positive) social interactions in larger cities.

We have added discussion of this topic in the paragraph quoted directly above on pages 15 and 16.

Related to this point, I noticed that one of the measures of segregation used was the Gini coefficient, which is traditionally a measure of wealth or income inequality.

While the Gini coefficient was developed (and is most often used) as a measure of wealth or income inequality, it is a statistic meant to measure inequality across groups (statistical distributions in populations), be they income classes, neighborhoods, or ages. It has been used expansively as a measure of inequality across different fields including: inequality in length of life, (Shkolnikov et. al., 2003), racial demographics (White, 1986), plant root hair density (He, et. al., 2005), and seasonal tourist demand (Fernandez-Morales, 2003). So, this measure can be and has been used to measure segregation as well income inequality.

I think that a brief explanation of each of the measures and why they were used would be helpful.

We have now added this to the methods section on pages 19-20:

Measures of Residential Segregation

As in our related work [32], all analyses were conducted across four different measures of residential segregation [73] in order to ensure that the results were not sensitive to any specific segregation metric. Each of these measures has its own drawbacks and benefits. Each one differs with respect to how changes in the spatial distribution of the population affect the measure and how the measure behaves with respect to an uneven distribution of population throughout the city.

These measures included the mean deviance measure:

$$\Delta_{g,i} = \frac{1}{M} \sum_m^M |N_{g,m,i}/N_{m,i} - N_{g,i}/N_i|,$$

with g indexing group, m indexing neighborhood, and i indexing city. This can be interpreted as the percentage of each group that would have to change residences to produce an even distribution throughout a city. However, the movement of people between neighborhoods that are above the mean for that group does not change this measure. In other words movement of individuals between two neighborhoods that have a higher percentage of black (or white) residents than the city as a whole will not

impact this measure. In addition, this measure does not account for cases in which some neighborhoods have a much larger share of the population.

The normalized segregation index:

$$D_{g,i} = \frac{\sum_m \left| \frac{N_{g,m,i}}{N_{m,i}} - \frac{N_{g,i}}{N_i} \right| \cdot N_{m,i}}{2 \cdot N_i \cdot \frac{N_{g,i}}{N_i} \cdot \left(1 - \frac{N_{g,i}}{N_i}\right)},$$

which is a normalized version of the mean deviance measure that takes into account the fact that different neighborhoods can have different population sizes.

The Gini Coefficient:

$$gini_{g,i} = \frac{\sum_m \sum_l \left| \frac{N_{g,m,i}}{N_{m,i}} - \frac{N_{g,l,i}}{N_{l,i}} \right| \cdot N_{m,i} \cdot N_{l,i}}{2 \cdot N_i^2 \cdot \frac{N_{g,i}}{N_i} \cdot \left(1 - \frac{N_{g,i}}{N_i}\right)},$$

which can be interpreted as measuring the proportion of individuals of the other group experienced by group g . Unlike the mean deviance measure it is sensitive to redistribution among neighborhoods above or below the population mean demographics.

Finally, the exposure Bgg index, also known as the correlation ratio (CR or η^2) or the mean squared deviation:

$$\eta_{g,i}^2 = \frac{\sum_m N_{g,m,i}^2}{N_{g,i} \cdot \left(1 - \frac{N_{g,i}}{N_i}\right)} - \frac{\frac{N_{g,i}}{N_i}}{1 - \frac{N_{g,i}}{N_i}}.$$

This measure attempts to capture the probabilities of random members of each group interacting given the demographic distribution. It accounts for both neighborhood size and the movement of individuals between neighborhoods above and below the mean.

4. This comment is about the accessibility of the paper. Overall, there was a lot of jargon, which seemed unnecessary and potentially confusing. For example, "heterophobia" in the context of bias can sound like a fear of heterosexual people. Why not use common terms (like segregation) throughout?

We have attempted to limit the technical jargon in the main text to what is necessary to convey our ideas and findings. In this specific case, “heterophobia” is well established in the sociology and network science literature that we draw from, e.g., see Rogers et. al., 1970; Lozares, et. al., 2014). Here, residential segregation is an observed variable that is distinct from heterophobia and is one potential cause of heterophobia (or reduced interactions with outgroup members).

Based on this comment we realized that we never defined scale-invariance and have added the follow text on page 4 (new text in bold):

*For example, in the case of average per-capita social interactions, k , the scaling relationship takes the form of $k \propto N^{\delta}$, where $\delta = 1/6$. **Here, scale-invariance refers to the property that doubling N results in a $2^{\delta} - 1 \approx 12\%$ increase in per-capita social interactions, k , regardless of initial values for N and k .***

5. It was unclear how the heat index was relevant in these analyses. I understand that past work found that to be a relevant factor predicting between-city differences in bias, but is there an explanation for why that is? How are the heat index and deprivation related?

Thank you for this question. Our point was that a “kitchen sink” approach in which many variables are thrown into a regression can miss the mechanism while still being predictive. (But it is also very likely to overfit).

As we discussed in our response to Reviewer #2, the heat index turned out to be highly correlated with city diversity, likely due to differences in agriculture and politics between northern and southern states. As a result, the authors of that paper found a spurious relationship between heat and implicit biases because when you control for city diversity the heat index effect goes away. Those authors also found a cluster of factors including the rate of premature death, total violent crimes, the rate of violent crimes, the rate of injury deaths, and the percentage of mental health providers as being predictors which were also related to implicit racial bias levels. These factors are generally associated with local deprivation which is why we chose to also include those variables in our analyses.

6. Please be careful with the use of causal language (e.g., "driving subsequent implicit biases")

Thank you for this suggestion, we have conducted additional temporal analyses and have toned down the causal language in the title and throughout the manuscript. We think that the new analyses have added a lot to the direction of the effects and their temporal manifestations.

We made the following changes to the text:

One pages 12 to 15 we have added an additional section:

Timescales of Temporal Precedence

The learning mechanism linking biases and between-group interactions emphasizes a specific causal direction in the model: interactions → bias levels. However, there are other mechanisms, such as selective migration [56] and individual mixing preferences, that may facilitate reverse causal pathways in which bias levels influence changes in diversity, population size, and segregation, respectively. While the heterophobia term in the model can account for the effects of mixing preferences (along with historical processes and explicit racism, e.g., that influenced unfair lending policies), our model does not explicitly account for processes in which implicit biases facilitate changes in city diversity and population size.

To begin to understand the role of each of these causal directions, we take advantage of the fact that 43 cities have implicit racial bias data available for all 10 years of data. We employ Granger causality [57] to statistically test whether changes in one variable precede or follow changes in another variable. In brief, these analyses test whether the linear regressions between two variables of interest improve when one of the variables is lagged in time (see Materials and Methods). We perform these analyses for each city and calculate the percentage of cities with statistically significant evidence of temporal precedence.

We find evidence that changes in population size, diversity, and segregation precede changes in implicit biases at a lag of one year for a majority of cities (Table 1, Figure 4). In contrast, only a fraction of cities show evidence for the reverse temporal precedence. Results are similar at a lag of 2 years. At a lag of 3 years, however, there is equal evidence for both temporal precedence directions.

In combination with the mathematical model presented here, these results suggest a mismatch in the timescales at which different mechanisms play out. In particular, these analyses suggest that, at short timescales (i.e., 1 - 2 years), changes in structural factors primarily precede changes in implicit racial biases as individuals learn from and internalize changing social contexts. However, there is also some evidence of the reverse temporal direction at these short timescales. This direction of bias changes preceding structural factors may be due to immediate, individual-level effects such as changes in bias levels leading to changes to individual mixing preferences.

At long timescales, evidence is present for influence in both directions from biases to structural factors and vice versa. This fits with our model's suggestion of rapid learning involved in setting implicit biases (Figure 3). Psychological adaptations to changing social conditions are expected to be faster than the speed with which individuals (and their households) can move to different neighborhoods or cities. Thus, we expect that changes in biases happen faster than changes in city demographics and patterns of segregation. More work is needed to enumerate potential mechanisms linking bias levels back to structural changes (e.g., selective migration [56]) and to mathematically model these mechanisms in an urban scaling context.

Figure 4: Granger causality analyses provide differing amounts of evidence for each direction temporal precedence across 43 cities. At lags of one and two years, more cities have evidence of changes in population preceding changes in bias. At a lag of three years, there is equal evidence for both directions. Error bars represent the bootstrapped standard error. The inset shows the same measure for diversity (dotted line) and segregation (dashed line) with Granger causality directions indicated by the same colors.

	1 year lag	2 year lag	3 year lag
population→bias	73.8 ± 7.0	78.6 ± 6.3	85.7 ± 5.5
bias→population	19.0 ± 5.9	35.7 ± 7.5	76.2 ± 6.2
diversity→bias	61.9 ± 7.3	66.7 ± 7.3	88.7 ± 4.9
bias→diversity	24.4 ± 6.7	45.2 ± 7.7	84.5 ± 5.6
segregation→bias	69.0 ± 7.1	76.2 ± 6.7	95.2 ± 3.3
bias→segregation	19.0 ± 6.3	38.1 ± 7.5	81.0 ± 5.9

In the discussion on page 15 (new text in bold):

The model developed here demonstrates that relatively simple considerations of heterogeneous mixing among a small number of social groups can explain a large proportion of why people in some cities have stronger implicit racial biases than in others. While it is somewhat surprising that only three factors - city population, majority group size, and racial segregation – account for so much between-city variance in bias levels, this is in line with recent evidence that implicit racial biases are driven more by social contexts than by individual differences in attitudes [58, 59]. Importantly, our

model and empirical evidence suggest that, at short timescales, implicit racial biases emerge from the interaction between city-wide social contexts that are shaped by the built environment and individual psychology, which determines how much and how quickly people learn from those contexts. At longer timescales, we find evidence that other, still unenumerated, and unmodeled mechanisms create feedback loops in which implicit racial biases shape these social contexts, e.g., through selective migration [56].

In the methods on page 22-23:

In order to evaluate evidence for temporal precedence between structural factors and implicit bias levels we employed Granger causality analyses [57] as implemented in the python statsmodels library. These tests start by fitting a linear regression of one of the three variables of interest (population, diversity, and segregation) and implicit bias levels for a single city using 10 years of data. Next, another linear regression is fit with one variable lagged in time against the other variable. Evidence that changes in the lagged variable preceded changes in the other variable is evaluated based on an F-statistic calculated by the percent change in the sum of squared residuals from the lagged model from the sum of squared residuals of the non-lagged model. This statistic is adjusted for the number of comparisons and the degrees of freedom to obtain an F-statistic and p-value. We conducted this analysis across all 43 cities with 10 years of data and for each choice of which variable to lag. We repeated this for lags of 1, 2, and 3 years.

To summarize the results, we computed the percentage of the 43 cities that show evidence ($p < .05$) for temporal precedence at each lag. Confidence intervals were computed by bootstrapping these percentages with replacement and computing the standard error. To combine evidence across the four segregation measures used, we averaged the percent for each measure and combined the standard errors according to:

$$\sigma_{combined} = \sqrt{\frac{\sum \sigma_i^2}{4}} \quad (12)$$

where σ_i are the standard errors computed for each segregation measure.

Thank you for your comments and suggestions.

Bibliography

Bettencourt, Luís MA, et al. "The interpretation of urban scaling analysis in time." Journal of the Royal Society Interface 17.163 (2020): 20190846.

Fernandez-Morales, Antonio. "Decomposing seasonal concentration." *Annals of tourism research* 30.4 (2003): 942-956.

He, Zhenxiang, et al. "Assessment of inequality of root hair density in *Arabidopsis thaliana* using the Gini coefficient: a close look at the effect of phosphorus and its interaction with ethylene." *Annals of botany* 95.2 (2005): 287-293.

Hugenberg, Kurt, and Galen V. Bodenhausen. "Facing prejudice: Implicit prejudice and the perception of facial threat." *Psychological Science* 14.6 (2003): 640-643.

Jacoby-Senghor, Drew S., Stacey Sinclair, and Colin Tucker Smith. "When bias binds: Effect of implicit outgroup bias on ingroup affiliation." *Journal of personality and social psychology* 109.3 (2015): 415.

Lozares, Carlos, et al. "Homophily and heterophily in personal networks. From mutual acquaintance to relationship intensity." *Quality & Quantity* 48 (2014): 2657-2670.

Rogers, Everett M., and Dilip K. Bhowmik. "Homophily-heterophily: Relational concepts for communication research." *Public opinion quarterly* 34.4 (1970): 523-538.

Shkolnikov, Vladimir M., Evgueni E. Andreev, and Alexander Z. Begun. "Gini coefficient as a life table function: Computation from discrete data, decomposition of differences and empirical examples." *Demographic Research* 8 (2003): 305-358.

White, Michael J. "Segregation and diversity measures in population distribution." *Population index* (1986): 198-221.

Wimmer, Andreas, and Kevin Lewis. "Beyond and below racial homophily: ERG models of a friendship network documented on Facebook." *American journal of sociology* 116.2 (2010): 583-642.

Reviewers' Comments:

Reviewer #1:

Remarks to the Author:

The authors have very carefully and thoughtfully addressed my comments, and I think the paper can be considered for publication. Great work!

Reviewer #2:

Remarks to the Author:

I am satisfied with the authors' responses to my previous review.

Reviewer #3:

Remarks to the Author:

This is a much clearer version of the manuscript. I appreciated the expanded explanations about the ceiling estimates, the additional details about the segregation measures, and the time-lagged analysis supporting a theoretical causal pathway. Even with time lagged analyses, though, I think the language should still avoid implying causality, but there can be an emphasis on how the associations are consistent with the theoretical account.

Overall, I think this manuscript makes an important contribution to understanding the ecological factors that may contribute to bias. I agree with the second reviewer in saying that this does not necessarily mean that people who move to larger cities will be less biased, but emphasizing that people should be aware of how structural factors could potentially impact bias is important.

Thank you and the reviewers for your comments which have greatly improved the manuscript.

Our response to reviewers are attached. We have also made the necessary changes to the manuscript in response to editorial comments.

Reviewer #1 (Remarks to the Author):

The authors have very carefully and thoughtfully addressed my comments, and I think the paper can be considered for publication. Great work!

Thank you for your helpful comments.

Reviewer #2 (Remarks to the Author):

I am satisfied with the authors' responses to my previous review.

Thank you for your helpful comments.

Reviewer #3 (Remarks to the Author):

This is a much clearer version of the manuscript. I appreciated the expanded explanations about the ceiling estimates, the additional details about the segregation measures, and the time-lagged analysis supporting a theoretical causal pathway. Even with time lagged analyses, though, I think the language should still avoid implying causality, but there can be an emphasis on how the associations are consistent with the theoretical account.

Overall, I think this manuscript makes an important contribution to understanding the ecological factors that may contribute to bias. I agree with the second reviewer in saying that this does not necessarily mean that people who move to larger cities will be less biased, but emphasizing that people should be aware of how structural factors could potentially impact bias is important.

We have further emphasized that we discuss causal implications of the work it is driven by the assumptions and mechanisms described in the mathematical model and not implications based on empirical data.

Particularly, the relevant sections of the discussion now read:

The model developed here demonstrates that relatively simple considerations of heterogeneous mixing among a small number of social groups can explain a large proportion of why people in some cities have stronger implicit racial biases than in others. While it is somewhat surprising that only three factors - city population, diversity, and racial segregation - account for so much between-city difference, this is in line with recent evidence that implicit racial biases are driven more by social contexts than by individual differences in attitudes [60, 61, 25, 26].

Our model provides a number of concrete theoretical predictions that can and should form the basis of new experimental hypotheses. First, our model predicts that at short timescales, implicit racial biases emerge from the interaction between city-wide

social contexts that are shaped by the built environment and individual psychology which determines how much and how quickly people learn from those contexts. We find preliminary support for this hypothesis by taking advantage of the longitudinal nature of our data (Figure 4, Table 1). At longer timescales, other still unenumerated, and unmodeled mechanisms may create feedback loops in which implicit racial biases shape these social contexts, e.g., through selective migration [58].

Second, our model implicitly predicts that on average, inter-group contact in cities is beneficial with respect to reducing implicit racial biases. This matches results from the urban scaling literature that includes psychological depression [62], economic outputs [63, 1, 2], and creative outputs [64]. In all of these cases, the observation of increasing beneficial returns to city population suggests that interactions across these modalities are, on average, positive. If this was not the case, we would expect to find all three main results reversed so that smaller, less diverse, and more segregated cities have lower bias levels. This was not what we found empirically.

However, the equations of urban scaling theory as formulated do not address interaction quality directly. There is likely great variation in interaction quality within cities. For example, inter-group contact may be cognitively costly [65], and interactions between individuals or in certain neighborhoods may be negative, particularly in areas with high levels of existing implicit racial biases [66]. Thus, investigations of whether and how cities systematically facilitate interactions of differing quality are natural next steps.

Finally, our model predicts that as more people move into cities over the next decades implicit biases may decrease so long as cities are not too segregated, remain centers of diversity, and residents learn from shifting social environments. In addition, our model predicts that decreasing segregation may lead to reductions in implicit racial biases that could have large societal impacts [67], though stronger causal evidence is needed to confirm these hypotheses. This is important to recognize, as cities with lower levels of racial segregation also tend to, not accidentally, have higher incomes [32] and healthier inhabitants [68].

Please do not hesitate to reach out with any additional questions or comments.